# Brominated VSLS and their influence on ozone under a changing climate

Stefanie Falk[1], Björn-Martin Sinnhuber[1], Gisèle Krysztofiak[2], Patrick Jöckel[3], Phoebe Graf[3], and Sinikka T. Lennartz[4]

[1]Institute of Meteorology and Climate Research, Karlsruhe Institute of Technology, Karlsruhe, Germany
[2]LPC2E, Université d'Orléans, CNRS, UMR7328, Orléans, France
[3]Deutsches Zentrum für Luft- und Raumfahrt e.V., Oberpfaffenhofen, Germany
[4]Geomar, Helmholtz-Centre for Ocean Research Kiel, Kiel, Germany

*Correspondence to:* Stefanie Falk (stefanie.falk@kit.edu)

**Abstract.** Very short-lived substances (VSLS) contribute as source gases significantly to the tropospheric and stratospheric bromine loading. At present, an estimated 25% of stratospheric bromine is of oceanic origin. In this study, we investigate how climate change may impact the ocean–atmosphere flux of brominated VSLS, their atmospheric transport, chemical transformations, and evaluate how these changes will affect stratospheric ozone over the 21st century.

Under the assumption of fixed ocean water concentrations and RCP6.0 scenario, we find an increase of the ocean–atmosphere flux of brominated VSLS of about 8–10% by the end of the 21st century compared to present day. A decrease in the tropospheric mixing ratios of VSLS and an increase in the lower stratosphere are attributed to changes in atmospheric chemistry and transport. Our model simulations reveal that this increase is counteracted by a corresponding reduction of inorganic bromine. Therefore the total amount of bromine from VSLS in the stratosphere will not be changed by an increase in upwelling. Part of the increase of VSLS in the tropical lower stratosphere results from an increase in the corresponding tropopause height. As the depletion of stratospheric ozone due to bromine depends also on the availability of chlorine, we find the impact of bromine on stratospheric ozone at the end of the 21st century reduced compared to present day. Thus, these studies highlight the different factors influencing the role of brominated VSLS in a future climate.

## 1 Introduction

Ozone is an important trace gas in the Earth's atmosphere. The stratospheric layer of its highest abundance, the ozone layer, absorbs harmful ultraviolet (UV) radiation threatening all lifeforms on the Earth's surface and acts as a potent greenhouse gas (GHG). In the troposphere, ozone is considered a harmful pollutant. Catalytic cycles involving bromine and mixed halogen reactions, namely with chlorine, efficiently deplete ozone (e.g., Sinnhuber et al., 2009). The ozone depletion efficiency of bromine is strongly related to the available amount of activated chlorine in the atmosphere (Yang et al., 2014; Sinnhuber and Meul, 2015; Oman et al., 2016). Long-lived, anthropogenically emitted, halogenated source gases (SG), e.g., $CH_3Br$ and Halons, have been restricted by the Montreal protocol and its amendments. Their atmospheric concentrations have started to decline globally (see Global Ozone Research and Monitoring Project, 2011, Chap. 1). Still, they contribute about 75% to the over-

all bromine loading in the stratosphere. The remainder is provided by organic SG of oceanic origin of which methyl bromide ($CH_3Br$), bromoform ($CHBr_3$), and dibromomethane ($CH_2Br_2$) are the most abundant. Minor brominated very short-lived substances (VSLS) include the mixed bromo-chloro-carbons $CHCl_2Br$, $CHClBr_2$, and $CH_2ClBr$. The tropospheric lifetime of these gases lies between several days to weeks. They are produced by plankton and macroalgae, and are predominantly

produced in coastal waters (Moore et al., 1996; Lin and Manley, 2012; Hughes et al., 2013; Stemmler et al., 2015). Through gas exchange governed by the concentration gradient between ocean water and atmosphere, solubility, and wind stress, VSLS are emitted into the atmosphere. Transport to the stratosphere, as shown by different model studies (Aschmann et al., 2009; Hossaini et al., 2012; Liang et al., 2014), occurs in tropical regions of deep convection, most importantly the Western Pacific and Maritime Continent, in South East Asia, and over the Gulf of Mexico. Organic SG are transported through the tropical

tropopause layer (TTL) together with their inorganic product gases (PG). PG are produced through photochemical decomposition of VSLS and provide reactive bromine ($Br_y$, from Br, $Br_2$, HBr, BrO, $BrONO_2$, $BrNO_2$, BrCl, and HOBr) to the stratosphere. This is schematically shown in Fig. 1. In recent years, several approaches have been taken to describe the stratospheric or regional abundance of bromine from VSLS. Top-down scenarios (Warwick et al., 2006; Liang et al., 2010; Ordonez et al., 2012) match atmospheric observations by setting constant fluxes or boundary concentrations. Bottom-up scenarios (e.g.,

Ziska et al., 2013) developed emission climatologies by extrapolating measurements in the surface ocean and marine boundary layer and calculate emissions accordingly. As shown by Lennartz et al. (2015), the bottom-up fluxes based on the oceanic water concentrations of Ziska et al. (2013) are in good agreement with available atmospheric VSLS observations. Recently, Ziska et al. (2017) have investigated the future evolution of the ocean–atmosphere fluxes of VSLS through the 21st century based on Coupled Model Intercomparison Project (CMIP) 5 model output and fixed atmospheric VSLS concentrations. They found

fluxes of $CH_2Br_2$ and $CHBr_3$ increasing by 6.4% (23.3%) and 9.0% (29.4%), respectively, dependent on the Representative Concentration Pathways (RCP) 2.6 (RCP8.0) scenario.

In this study, we will address the open questions on how these oceanic emissions of VSLS evolve in response to a changing climate and changing atmospheric concentrations (Section 3), how transport and tropospheric chemistry influence the stratospheric bromine abundance in a changing climate (Section 4), and how stratospheric ozone will be affected by the assumed

changes in VSLS abundance (Section 5). Details about the model and simulations used in this study will be given in Section 2.

## 2 Model and experiments

All model experiments have been performed using the ECHAM/MESSy Atmospheric Chemistry (EMAC) model (Jöckel et al., 2010). Table 1 gives an overview over the key-factors of the simulations.

Future changes in fluxes of brominated VSLS from the ocean are studied with a free-running long-term simulation (SC_free, 1979–2100) using a simplified chemistry (Section 3), augmented by a similar simulation, but nudged towards the European Centre for Medium-Range Weather Forecasts (ECMWF) Re-Analysis (ERA)–Interim over the period 1979–2012. Therein, VSLS emission fluxes are computed online from prescribed sea water concentrations. The simplified chemistry simulations

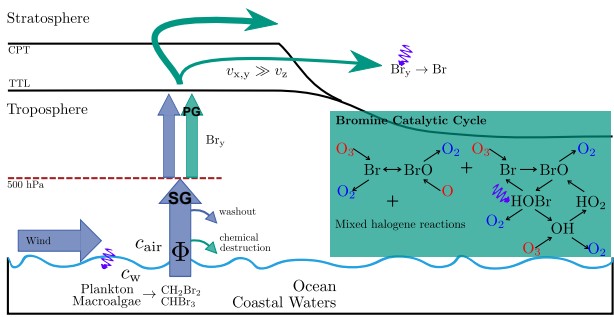

**Figure 1.** Scheme of VSLS emission and catalytic cycle of ozone depletion involving bromine. (left) VSLS are produced by plankton and macroalgae predominantly in coastal waters. They are emitted through gas exchange between ocean ocean and atmosphere. These organic source gases (SG) undergo chemical transformation into inorganic product gases (PG). Both are convectively transported through the tropical tropopause layer (TTL). Through photochemical decomposition, reactive bromine $Br_y$ is provided to the stratosphere. (right) Two examples of catalytic cycles of ozone depletion involving bromine. A + indicates increasing order of catalytic complexity. Reactants are shown in red, catalysts in black, and products in blue. Photochemical reactions are indicated by a violet wave.

use EMAC version 2.50 with submodels `airsea` (Pozzer et al., 2006) (with $k_w = 0.222\,u^2 + 0.333\,u$ parametrization quadratic with respect to wind speed (Nightingale et al., 2000)), `cloud`, `cloudopt`, `convect` (with operational ECMWF convection scheme), `cvtrans`, `ddep` (Kerkweg et al., 2006), `ptrac` (Jöckel et al., 2008), `rad`, `scav` (Tost et al., 2006), `surface`, and `tnudge` (Kerkweg et al., 2006). The set-up is as in Lennartz et al. (2015); Hossaini et al. (2016). Chemical reactions are

not computed interactively, e.g., via the EMAC submodul `mecca` (Sander et al., 2011a). The VSLS lifetime due to reaction with OH has been fixed to monthly mean values from the National Centre for Meteorological Research (CNRM) (Michou et al., 2011; Morgenstern et al., 2016) model calculations, while photolysis rates are computed within the EMAC submodel `jval` (Sander et al., 2014). Only in these simulations with simplified chemistry, OH concentrations have been set to zero in the lower troposphere (700–1000 hPa) to reduce the variability of ground level volume mixing ratio (VMR) of VSLS. The

chemical lifetime of VSLS in the lower troposphere is therefore overestimated. Due to the longer lifetime, VSLS are more abundant in the lower troposphere leading to a flux suppression. Water concentrations of $CH_2Br_2$ and $CHBr_3$ have been held constant using the climatology of Ziska et al. (2013). For mixed bromo-chloro-carbons ($CHBrCl_2$, $CHBr_2Cl$, $CH_2BrCl$), water concentrations have been estimated by scaling $CHBr_3$ concentrations to obtain a better agreement between model simulation and tropical mean profile and surface observations of these VSLS. Based on the lifetime estimate, VSLS are decomposed

and converted to $Br_y$. The partitioning of $Br_y$ into Br, $Br_2$, HBr, BrO, $BrONO_2$, $BrNO_2$, BrCl, and HOBr in these simplified chemistry simulations has been computed offline from a full-chemistry EMAC simulation of one year duration with six hourly output. Scavenging is applied to Br, $Br_2$, HBr, $BrNO_2$, $BrONO_2$, BrCl, and HOBr. Concentrations of $CO_2$, $CH_4$, CFC, and $N_2O$ in SC_free are taken from a CNRM CM5 model (Voldoire et al., 2013) simulation with RCP6.0 scenario (Fujino et al., 2006; Hijioka et al., 2008).

Data of a full-chemistry long-term simulation (RC2-base-05, Jöckel et al., 2016) over a time span of 150 years (1950–2100)

and performed as part of a Chemistry-Climate Model Initiative (CCMI) recommended set of simulations by the Earth System Chemistry-Climate Modelling (ESCiMo) consortium will be used for studying changes in transport and photochemical transformation of bromine species (Section 4). In this simulation, VSLS fluxes have been held constant following scenario five of Warwick et al. (2006).

An intermediate-term experiment, consisting of a set of two simulations and spanning the years 2075–2100, has been performed for assessing implications on ozone depletion in a future climate with significantly lower chlorine loading in the atmosphere (Section 5). The simulations named RT1a and RT1b both include online computation of aerosol formation. Fluxes of $CH_2Br_2$ and $CHBr_3$ are computed online from sea water concentrations of Ziska et al. (2013) using the EMAC submodel `airsea` as in RC2-base-05 with the $k_w$ parametrization according to Wanninkhof (1992), which is strictly quadratic with respect to wind

speed ($k_w = 0.31\,u^2$). For assessing the impact of VSLS on ozone, all VSLS emissions have been switched off in RT1b. The impact of various $k_w$ parametrizations on VSLS emission has been previously studied. The differences on the global level are <15% (cf. Tab. 4 in Lennartz et al. (2015) when comparing Nightingale et al. (2000) and Wanninkhof and McGillis (1999)). For wind speeds exceeding $10\,\mathrm{ms}^{-1}$ the Wanninkhof (1992) $k_w$ parametrization diverges slightly stronger towards higher transfer velocities compared to the Nightingale et al. (2000) parametrization (cf. Fig. 1 in Wanninkhof and McGillis (1999) and Fig. 2

in Lennartz et al. (2015)). Regarding integrated global emissions of VSLS, both parametrizations result in similar fluxes, given that the mean global wind speed lies in a range where these parameterizations do not differ drastically. However, the Nightingale et al. (2000) parameterization reacts more sensitive to changes in wind speed, which introduces a further uncertainty when assessing changes over time in a changing climate. The full-chemistry experiments use EMAC version 2.51 (RC2-base-05) and 2.52 (RT1a/b), respectively. The dynamics have not been specified except for a weak nudging of the equatorial wind

quasi-biennial oscillation (QBO). RC2-base-05 combines hindcast with future projections. The set-up of RT1a and RT1b is almost identical to RC2-base-05, thus we refer to the corresponding paper by the ESCiMo consortium (Jöckel et al., 2016) for general information. The major difference lies in the aforementioned treatment of VSLS emission, which is handled analogous to SC_free, except for mixed bromo-chloro-carbons emissions taken from Warwick et al. (2006). Since heterogeneous reaction and chlorine activation are important for the depletion process of ozone, tropospheric and stratospheric aerosol formation is

computed online using the submodel `GMXe` (Pringle et al., 2010) of EMAC. The set-up has been adapted from RC1-aero-07 (Jöckel et al., 2016) with modifications as described by Brühl et al. (2012, 2015). Radiation coupling had been activated in `GMXe`, but cloud coupling had not been activated. In this regard, an additional oceanic sulfur source, carbonyl sulfide (COS), which is a major source of stratospheric sulfur has been included in addition to dimethyl sulfide (DMS). Whereas the emission of the latter is computed from prescribed ocean concentrations, constant fluxes of COS have been adopted from Kettle et al.

(2002). Additional reaction pathways of sulfur have been enabled accordingly. RT1a and RT1b have been initialized with available monthly mean values from RC2-base-05. COS has been initialized from a simulation which results have been published recently (Glatthor et al., 2015), including an artificially increased oceanic source to close the atmospheric budget.

The model's spatial resolution is T42L39MA for the simplified chemistry experiments, T42L47MA for RC2-base-05, and T42L90MA for RT1a/RT1b, respectively, corresponding to a $2.8° \times 2.8°$ grid, with a top level at $0.01\,\mathrm{hPa}$, and 39, 47, or 90

vertical hybrid-pressure levels. The mean tropical troposphere (below $100\,\mathrm{hPa}$) is discretised into 16, 26, or 27 levels, and the

**Table 1.** EMAC model experiments used in this study. All experiments follow the RCP6.0 scenario of GHG emissions and have accordingly prescribed SST and SIC from HadGEM2.

| Experiment | Model Version | Resolution | Time-Span | Chemistry | VSLS Emission | Interactive Aerosols |
|---|---|---|---|---|---|---|
| SC_nudged | 2.50 | T42L39MA | 1979–2012 | simplified bromine | `airsea` | no |
| SC_free | 2.50 | T42L39MA | 1979–2100 | simplified bromine | `airsea` | no |
| RC2-base-05 | 2.51 | T42L47MA | 1950–2100 | full | Warwick et al. (2006) | no |
| RT1a | 2.52 | T42L90MA | 2075–2100 | full + sulfur | `airsea` | yes |
| RT1b | 2.52 | T42L90MA | 2075–2100 | full + sulfur | none | yes |

mean tropical stratosphere between $100\,\mathrm{hPa}$ and $1\,\mathrm{hPa}$ consists of 15, 15, or 48 levels, respectively. Emissions of GHG follow the RCP6.0 scenario and sea surface temperature (SST) and sea ice cover (SIC) are prescribed from Hadley Centre Global Environment Model version 2 (HadGEM2) forced with the RCP6.0 scenario for all simulations accordingly.

## 3  Long-term Trends in Oceanic Emission Fluxes

In this section, we investigate how a changing climate may influence emission fluxes of VSLS from the ocean. We will assess the impact of changing physical factors (e.g., SST, SIC, and wind speed) on ocean–atmosphere gas exchange driven by the RCP6.0 scenario. Here we assume constant oceanic concentrations of VSLS over the course of the century (following Ziska et al. (2013); Lennartz et al. (2015)). This specific assumption might not hold since the effects of climate change, e.g., increase of ocean temperature, acidification, change of salinity, and nutrient input, on marine organisms and thus the production of $CH_2Br_2$ and $CHBr_3$ is not yet fully understood. Recent combined marine ecosystem model studies imply a global decrease of net primary production (NPP) by plankton over the course of the 21st century (Laufkötter et al., 2015, 2016). However, the impact on bromocarbon concentration, predominantly produced by macroalgae in coastal regions, remains unclear.

As implemented in the EMAC submodel `airsea` (Pozzer et al., 2006), the flux of a gas dissolved in ocean water to the atmosphere is governed by its concentration gradient $\Delta c$ and transfer velocity $k$,

$$\Phi = k \cdot \Delta c \tag{1}$$
$$= k \cdot (c_\mathrm{w} - H \cdot c_\mathrm{air}),$$

with $k = (1/k_\mathrm{w} + R \cdot H \cdot T_\mathrm{air}/k_\mathrm{air})^{-1}$, wherein, $R$ is the universal gas constant and $H$ is the Henry coefficient for a specific gas. The transfer velocity depends largely on air temperature $T_\mathrm{air}$ and surface wind speed, which is taken into account by distinguishing between water- and air-side transport velocities ($k_\mathrm{w}$, $k_\mathrm{air}$). $k_\mathrm{w}$ is a polynomial function of wind speed depending on the chosen parametrization as mentioned in Section 2. The corresponding water and atmospheric concentrations are named $c_\mathrm{w}$ and $c_\mathrm{air}$.

In Fig. 2, the difference of VSLS fluxes with respect to the start of the simulation in 1979 is shown for the free-running and

**Table 2.** Average absolute flux for the year 2000 in $\mathrm{Gg\,yr^{-1}}$ and percentage of relative increase in VSLS flux between 2000 and 2100 from SC_free. The numbers have been obtained by linear regression of the data shown in Fig. 2 and evaluated at the given years.

| Region | $CH_2Br_2$ | | $CHBr_3$ | |
|---|---|---|---|---|
| | $(\mathrm{Gg\,yr^{-1}})$ | (%) | $(\mathrm{Gg\,yr^{-1}})$ | (%) |
| 90°N–50°N | 0.6 | 54.6 | 23.5 | 25.0 |
| 50°N–20°N | 8.2 | 14.6 | 41.7 | 8.7 |
| 20°N–0 | 19.2 | 6.6 | 52.4 | 8.6 |
| 0–20°S | 5.5 | 11.9 | 44.4 | 10.0 |
| 20°S–50°S | 4.3 | 18.0 | 33.2 | 8.2 |
| 50°S–90°S | 6.9 | 6.8 | 12.7 | 8.9 |

nudged simplified chemistry simulation. For both, $CH_2Br_2$ and $CHBr_3$, all zonal bands display linearly rising fluxes. The strongest increase with respect to 1979 values is found in the tropical zone (20° N–20° S) with roughly $2.5\,\mathrm{Gg\,yr^{-1}}$ and $13\,\mathrm{Gg\,yr^{-1}}$ for $CHBr_3$ and $CH_2Br_2$ respectively. Relative to the absolute value of the zonally averaged fluxes, this yields an increase of about 10% over the course of the century (Table 2). The increase is slightly stronger in the southern tropics. The strongest relative increase in flux is found in the northern hemisphere polar region (90° N–50° N), with 25% and roughly 55% for $CHBr_3$ and $CH_2Br_2$, respectively.

 Regarding the changing physical factors, the HadGEM2 prescribed SSTs are increasing almost linearly over the course of the century (Fig. 3a). Under the RCP6.0 scenario, this increase in SST ranges between $1\,°\mathrm{C}$–$3.5\,°\mathrm{C}$. The weak rise in Antarctic SST is accompanied by a weakly increasing Antarctic flux of VSLS. The corresponding HadGEM2 prognosticated retreat of Arctic sea ice is shown in Fig. 3b. Sea ice is not regarded as a source of VSLS in our study and therefore only acts as a lid blocking the ocean to atmosphere flux. Since the water concentrations from Ziska et al. (2013) used in our simulations do not take SIC into account, water concentrations have been extrapolated for regions typically covered by ice at present. In the `airsea` submodel, if SIC (fraction of grid box) is larger than 0.5, the transfer velocity ($k_\mathrm{w}$) is equal to zero, in other cases $k_\mathrm{w}$ is scaled depending on the fraction of SIC. Hence, a polar sea which is to a large extent free of sea ice has increased fluxes of VSLS in our future simulations. However, there are large uncertainties regarding the VSLS water concentrations in the future polar sea, for the polar ecosystem as a whole is undergoing a drastic change. In accordance to the general increase in flux, the Arctic August–September maximum of flux is expected to be more pronounced. On both hemispheres, seasonal cycles in zonally averaged VSLS fluxes peak in the summer months and show minima in late winter. There is a slightly stronger increase of fluxes in the future during the time periods of maxima, but no change in phase. Negative emissions representing a net sink of atmospheric $CH_2Br_2$ are found during winter at high-latitudes on the northern hemisphere. In the northern tropics, $CHBr_3$ shows a distinct maximum in northern hemisphere summer, while the southern tropics do not display any seasonal cycle. Albeit increased ocean–atmosphere fluxes in the future, only taking the changes of physical factors into account, seasonal cycles remain largely the same. In our simulations, zonally averaged absolute wind speed at $10\,\mathrm{m}$ is only slightly changing

over the course of the 21st century and with varying sign (-4–2%). Thus, it is indicated by our simulations but not explicitly shown, that the important factor regarding an increase of ocean–atmosphere flux of VSLS is the change in SSTs.

In Fig. 4, resulting VMR profiles of organic ($Br_{org}$) and inorganic bromine ($Br_y$) from VSLS are shown. VSLS data have been averaged over a time period 1990–2000 and 2090–2100. The VMR profile of $Br_{org}$ displays a steep decline in the lower troposphere (400–1000 hPa) by more than 50% of ground level VMR, stays almost constant before entering the stratosphere, where VSLS are quickly dissociated. Comparing present day and future values (keeping in mind that OH concentrations are nudged towards monthly mean values of OH and photolysis rates are fixed in SC_free), $Br_{org}$ is found to have increased throughout the atmosphere by about 0.1–0.4 ppt. Surface values of VSLS increase by 0.47 ppt (9%), while in the lower stratosphere, the increase amounts to 0.3 ppt (8%). VMR of $Br_y$ is increased from the lower stratosphere upwards by roughly 0.4 ppt (10%). These changes in the vertical profiles can be attributed to enhanced emissions in a future climate which, as shown, are of the order of 10% in the tropics.

As can be inferred from Eq. (1), ocean–atmosphere fluxes are sensitive to the abundance of VSLS in the atmosphere as well as a differing wind speed parametrization. An increased chemical dissociation of VSLS in the lowermost troposphere (e.g., due to a probable future increase in OH) would reduce the atmospheric concentration and therefore increase the flux from the ocean to the atmosphere without necessarily increasing the actual amount of bromine which is transported to the stratosphere. The total amount of bromine from VSLS transported through the UTLS strongly depends on the washout of inorganic PG ($Br_y^{VSLS}$) and hence on the partitioning and heterogeneous reactions converting $Br_y^{VSLS}$ between soluble, e.g., HBr, HOBr, and insoluble, e.g., BrO, species (e.g. Aschmann et al., 2009; Liang et al., 2014). Since OH concentrations in the lower troposphere have been set to zero in SC_free, the atmospheric lifetime and the resulting abundance of VSLS in the lower troposphere is enhanced. Therefore, the total ocean–atmosphere flux is suppressed. In this regard, fluxes from RT1a at the end of the 21st century have been compared to SC_free within the same time period. Much stronger fluxes (1.3–1.5 times) have been found in RT1a in comparison to SC_free. Particularly, no net sink for $CH_2Br_2$ occurs at high latitudes in RT1a. This partly explains the smaller increase in comparison to results recently published by Ziska et al. (2017). Ziska et al. (2017) diagnosed the flux from parameters such as SST and wind speed for a fixed VSLS concentration gradient and for different CMIP5 model simulations. They found an increase in flux of $CHBr_3$/$CH_2Br_2$ of 29.4%/23.3% for the RCP8.0 scenario and 9.0%/6.4% for RCP2.6, respectively. In addition to the smaller absolute fluxes due to the artificial suppression caused by setting OH to zero in the lower troposphere, we expect a smaller increase in flux from a theoretical point, taking Eq. 1 in to account, since we allow atmospheric concentrations to respond to changing flux. Underlying changes in photochemical dissociation and tracer transport due to a changing climate have not been disentangled at this point and will be studied in detail in the following section.

## 4 Stratospheric bromine loading

In addition to the possible increase in oceanic VSLS emissions due to climate change, discussed in the previous section, atmospheric transport and chemical transformation processes are also sensitive to climate change and may contribute to a change in the future stratospheric bromine loading from VSLS. These aspects will be studied in this section, based on the RC2-base-05

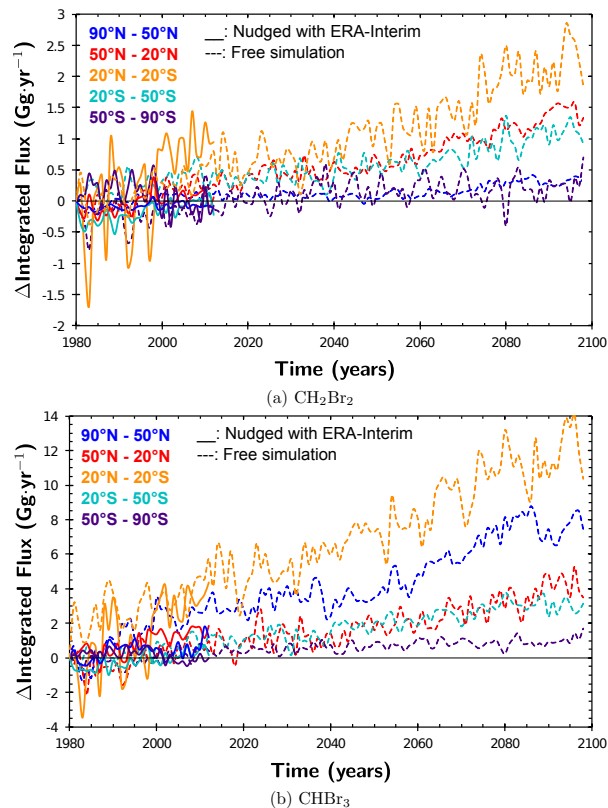

(a) $CH_2Br_2$

(b) $CHBr_3$

**Figure 2.** Difference of integrated flux separated in different zonal bands. Simplified chemistry EMAC simulation (SC_free and SC_nudged) with `airsea` gas exchange, water concentrations held constant. Solid lines represent ERA–Interim nudged (1979–2012) and dashed lines free-running (1979–2100).

ESCiMo simulation, spanning 150 years from 1950 to 2100, assuming constant VSLS fluxes. Hence the fluxes of VSLS do not response to changes in the ground level abundances of VSLS.

In Fig. 5, profiles of brominated substances are shown for the tropics. The profiles are weighted by the amount of bromine atoms per molecule. The whole 150 year data set has been smoothed using a moving average with a box window size of 11 years to account for, e.g., seasonal variations, and the solar cycle. From these smoothed data, three reference years have been chosen for the analysis: 1980, 2016, and 2100. Therefore, 2100 is referring to June of the last valid year of the smoothed data (2094). To guide the eye, the corresponding mean tropical tropopause heights from the model output are shown together with the profiles. There is an upward shift of the tropopause height of about $8\,\mathrm{hPa}$ between present day and future. An upward shift of VSLS VMR profiles in 2100 in comparison to past/present day profiles is also visible. In the RCP6.0 scenario, ground level VMR of $CH_3Br$ and VSLS are constant from 2016 onward. In case of $CH_3Br$, this roughly amounts to 1980 values. For all years, we find a fast decrease of VSLS of $5\,\mathrm{ppt}$ with a standard deviation of $0.25\,\mathrm{ppt}$ (or about 50% compared to ground level

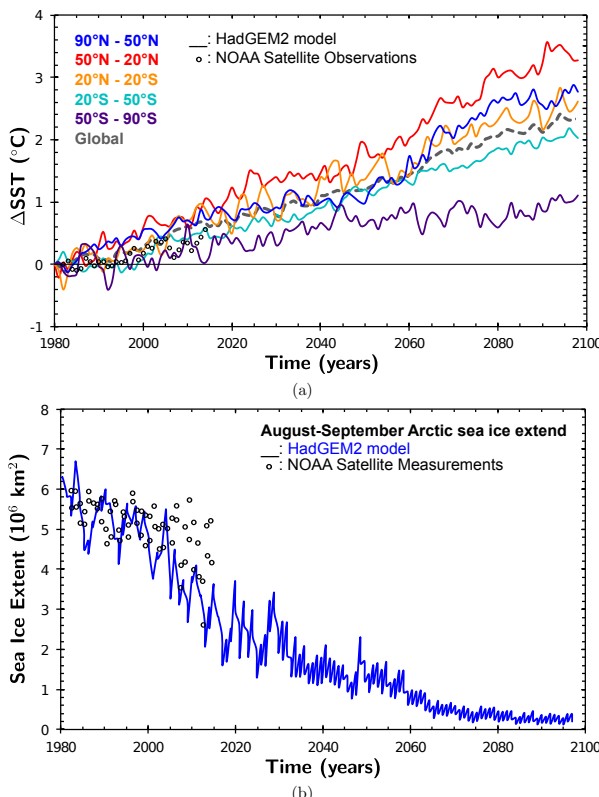

**Figure 3.** HadGEM2 prescribed ocean properties in the simplified chemistry simulations compared to National Oceanic and NOAA Optimum Interpolation (OI) V2 fields (Reynolds et al., 2002). (a) Change in sea surface temperature for different latitude bands. Global average is shown as dashed gray line. (b) Arctic sea ice extend in August and September.

VMR) between the surface and the mid-troposphere at about $500\,\mathrm{hPa}$. The comparison of the difference of profiles between future and past/present (Fig. 5a, lower panel) reveals decreasing bromine values from VSLS by about $0.1$–$0.8\,\mathrm{ppt}$ throughout the troposphere, while there is an increase of the same order of magnitude in the lower stratosphere. Similar results have been published for RCP4.5 and RCP8.5 scenarios, attributing these to changes in the tropospheric circulation and to the primary

5   oxidant OH (Hossaini et al., 2012). The amount of inorganic PG from VSLS ($\mathrm{Br}_y^{\mathrm{VSLS}}$) in the UTLS is decreasing by the same order of magnitude due to the enhanced upwelling in the tropics. As air in the UTLS becomes younger in a future climate, less SG ($\mathrm{Br}_{\mathrm{org}}^{\mathrm{VSLS}}$) will be dissociated into PG ($\mathrm{Br}_y^{\mathrm{VSLS}}$) compared to present day. For 2016, this decline is compatible with a decreasing amount of VSLS in the troposphere. A slight excess of $\mathrm{Br}_y^{\mathrm{VSLS}}$ in the stratosphere is found for 1980 in comparison to 2016 and 2100. This excess and the strong variability (denoted by the shown standard deviation) can be attributed to

10   the hindcast period (1950–2005) of the simulation including volcanic eruptions. Large volcanic eruptions can influence the transport of bromine from VSLS into the stratosphere which may be related to a similar effect as seen in stratospheric water

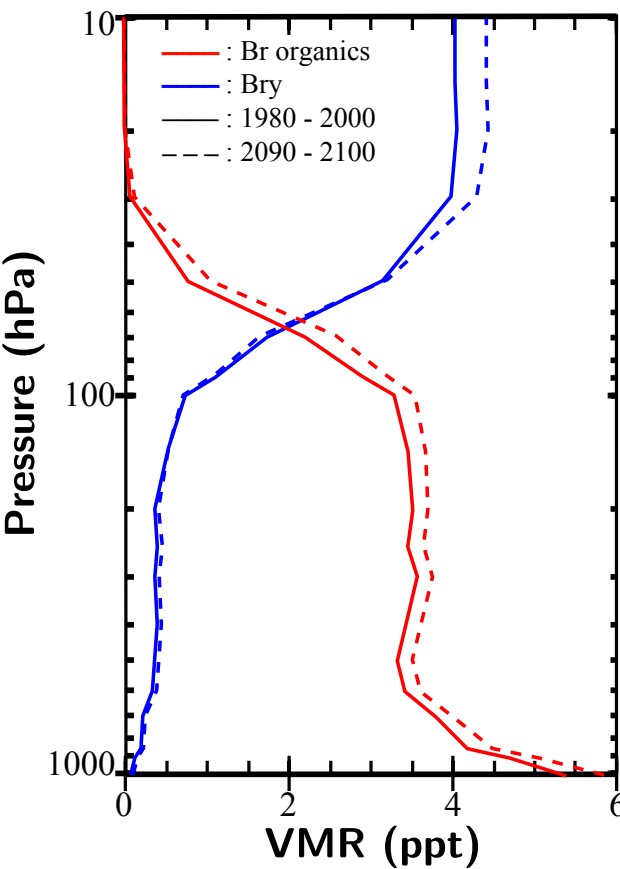

**Figure 4.** Tropical zonal mean (20° N–20° S), temporally averaged, vertical profiles of organic ($Br_{org}$) and inorganic bromine ($Br_y$) for the time periods 1990–2000 and 2090–2100 from SC_free. In consistency with increasing VSLS fluxes by 10% in the tropics, an increase of roughly 10% in $Br_y$ from VSLS in the stratosphere is found, while the increase in $Br_{org}$ amounts to 8%.

vapor (Löffler et al., 2016). Since volcanic activity has not been included in the future scenario, there is no such impact on $Br_y^{VSLS}$ from 2005 onward.

The largest change between present and future stems from the estimated decrease of long-lived SG, in particular Halons and $CH_3Br$. At present, Halons contribute about 6–7 ppt to the total bromine loading of the lower stratosphere ($\sim$23 ppt), which

5    is about the same amount as VSLS and $CH_3Br$ in RC2-base-05, whereas by the end of the century their contribution is reduced significantly to 1–2 ppt of total bromine ($\sim$17 ppt). This decline in long-lived, anthropogenically emitted SG is altering the amount of bromine released in the stratosphere on longer time scales. VSLS are already reduced due to photochemical dissociation when entering the TTL, while Halons are dissociating more slowly, providing a long lasting source of bromine to the stratosphere (Fig. 5b, lower panel). It is important to note, that although there is an increase of $Br_{org}^{VSLS}$ of 0.5 ppt in

10    the stratosphere assuming constant ocean–atmosphere fluxes, the overall amount of bromine in the stratosphere due to VSLS ($Br_{tot}^{VSLS}$) might be decreasing in the future. This depends on whether PG ($Br_y^{VSLS}$) are transported alongside the VSLS into

the UTLS or removed through washout in the troposphere. The model representations of underlying processes, e.g., conversion between soluble and insoluble inorganic bromine species through heterogeneous chemical reactions, are still uncertain.

In the following, we will derive a semi-analytic model to separate various aspects affecting the future VSLS distribution in the atmosphere (Section 4.1). Since the atmospheric window for air entering the stratosphere is located in the tropics, we will focus on averaged tropical atmospheric quantities. Subsequently, the transition between troposphere and stratosphere caused by a rising tropopause is influencing the interpretation of VMR profile differences between present and future. This will be discussed in Section 4.2.

## 4.1 Quantification of future atmospheric changes affecting VSLS mixing ratio profiles

The increase of VSLS in the stratosphere in the future can be attributed to changes in chemical and photolytical dissociation rates, and alternating transport from source regions through the TTL caused by a speed-up of the Brewer–Dobson circulation (BDC) (Hossaini et al., 2012). All of these factors influence the lifetime of VSLS in the atmosphere. A volume of air in a certain height (or rather pressure coordinate) shall have an associated mean temperature $T$, OH concentration [OH], photolysis frequency $J$, and age of air (AOA). In the model, VSLS are dissociated photochemically via

$$CH_2Br_2 + OH \rightarrow 2Br + H_2O, \tag{R1}$$

$$CHBr_3 + OH \rightarrow 3Br + H_2O, \tag{R2}$$

and

$$CH_2Br_2 + h\nu \rightarrow 2Br + products, \tag{R3}$$

$$CHBr_3 + h\nu \rightarrow 3Br + products. \tag{R4}$$

The simplification in these equations compared to reality is justified since the intermediate reaction products $CBr_2O$ and $CHBrO$ insignificantly amount to the total bromine PG (Hossaini et al., 2010). From the resulting first order differential equation

$$\frac{d[A]}{dt} = -k_A(T) \cdot [OH] \cdot [A] - J_A \cdot [A], \tag{2}$$

with [A] any of the VSLS species concentration, $k_A(T)$ the temperature dependent rate coefficient, $J_A$ the photolysis frequency (Sander et al., 2011b), and assuming [OH] is unchanged by the reaction, a simple solution of Eq. (2) is derived

$$[A](t) = \exp\left(-(k_A(T) \cdot [OH] + J_A) \cdot (t - t_0)\right) \cdot [A]_0. \tag{3}$$

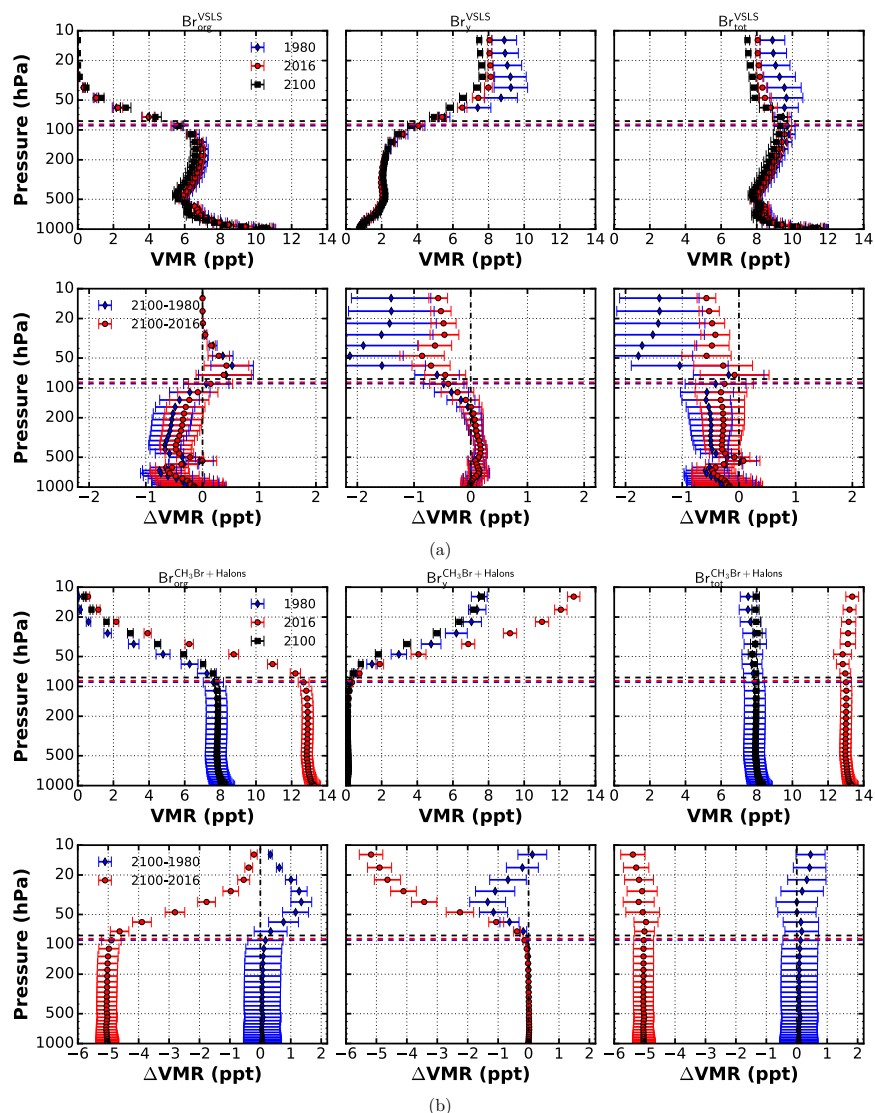

**Figure 5.** Vertical profiles of brominated substances divided into SG ($Br_{org}$), PG ($Br_y$), and SG + PG ($Br_{tot}$) in the tropics ($20°$ N–$20°$ S). Data from ESCiMo RC2-base-05 simulation (Jöckel et al., 2016). Absolute values of VMR in upper panel, difference $\Delta$VMR with respect to 2100 values in lower panel. (a) Bromine from VSLS; (b) Bromine from $CH_3Br$ and Halons.

Based on above equation, the influence of [OH], temperature, transport, and photolysis rate can be studied. For inferring the change in chemical dissociation, 10 year average profiles of [OH], and one year average profiles of temperature have been computed from RC2-base-05 data for present day (2016) and future (2100). The idea is to assess the effect of transport timescales ($t_0 \rightarrow t$) by using 10 year averages of mean AOA from RC2-base-05 data (neglecting the age spectrum in the

described volume of air). AOA shall refer to the time since an air parcel has been in contact with ground level. It has been evaluated in the EMAC simulation from an artificial, passive tracer with linearly increasing emission. Photolysis frequencies have been computed from averaged tropical profiles of temperature, humidity, and ozone column using the column version of `jval` (Sander et al., 2014) from EMAC. In case of photolysis frequencies, temperature dependence will not be discussed

separately.

In Fig. 6a, averaged profiles of temperature, AOA, and [OH] are shown. Mean tropospheric temperatures are higher, while stratospheric temperatures are lower in the future, which is in line with other studies (e.g., IPCC - Intergovernmental Panel on Climate Change (2013, Chap. 12), Global Ozone Research and Monitoring Project (2014, Chap.2)). The concentration of OH is increased throughout the atmosphere, apart from the lowermost levels. AOA will become notably younger within the

stratosphere by the end of the century, as shown in various other studies (Austin et al., 2007, 2013; Butchart et al., 2006; Li et al., 2008; Muthers et al., 2016), and become slightly older (by a few days) in the troposphere. Vertical profiles of the $CHBr_3$ lifetime for present and future are shown in Fig. 6b. Because of an increase in photolysis rates due to increasing temperatures, the $CHBr_3$ lifetime is decreasing. $CH_2Br_2$ is not shown, since its lifetime with respect to photolysis is almost infinite in the troposphere and thus determined by reaction with OH.

By varying the variables in Eq. (3) one by one, the impact of each on the resulting profile difference $(\Delta[A](t)/[A]_0)$ has been calculated (Fig. 7). The increase of [OH] in the RCP6.0 scenario results in a general decreasing VMR of VSLS in the troposphere and lower stratosphere. The influence is highest at $500\,\mathrm{hPa}$ and around the tropopause. $CH_2Br_2$ is affected more strongly by a change of [OH] due to chemical destruction ($\sim$5%) than $CHBr_3$ ($\sim$2%). The change in mean temperatures causes a tropospheric VMR decrease by at most 2%. In the stratosphere decreasing temperatures increase $\Delta[A](t)/[A]_0$ by

about 1%. An increased AOA in the troposphere is reflected by a decreasing $\Delta[A](t)/[A]_0$ ($\sim$2%), while the opposite is true for the juvenescence of air in the stratosphere (8–18%). For $CH_2Br_2$, the impact of AOA is apparently overestimated in the lower stratosphere by this ansatz, which might be because of the neglected AOA spectrum representing a mixing of different air masses. Since the photolytical lifetime of $CH_2Br_2$ in the troposphere is infinite, it has no influence on the tropospheric part of the profile. A weak decrease in the order of 1–2% is apparent in the lower stratosphere. This has been found to be mainly

driven by temperature sensitivity of photolysis. In case of $CHBr_3$, a 1–2% decrease due to changing photolysis is found in the free and upper troposphere. This change in photolysis rate is mainly due to changes in tropical ozone abundance.

If all occurring changes are included, the actual profile differences between future and present are rather well reproduced (shown in red). These VMR profiles are 10 year averages for the tropics. The corresponding standard deviation is plotted as shaded error band. The decreasing VMR of VSLS in the troposphere is in the order of 5–7% (at about $250\,\mathrm{hPa}$) for $CH_2Br_2$

and $CHBr_3$, respectively. In the stratosphere, the maximum increase, dependent on specie, occurs at differing pressure levels and amounts to 7–8%.

In Summary, all occurring future changes are decreasing VMR of VSLS in the troposphere. In case of $CHBr_3$, all factors are of the same order of magnitude. The tropospheric decrease of $CH_2Br_2$ VMR is mainly driven by increasing [OH]. In the upper troposphere / lower stratosphere (UTLS), the impact of the juvenescence of AOA dominates, inflicting an increase in VMR.

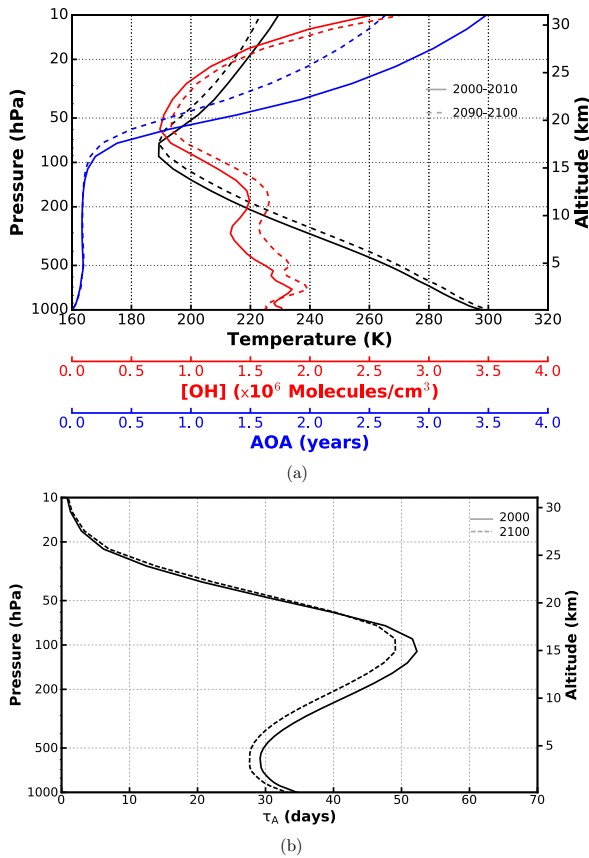

**Figure 6.** Tropical average profiles from ESCiMo RC2-base-05 simulation of changing variables in Eq. (3) for present day (solid lines) and future (dashed lines). (a) Temperature, OH concentration, and age of air; (b) Lifetime of $CHBr_3$.

## 4.2 Implications of a rising tropopause on VSLS mixing ratio profiles

The GHG induced warming of the troposphere and cooling of the stratosphere causes a rise of the tropopause. Model mean tropical tropopause heights from RC2-base-05 have been smoothed using a moving average with a box window size of 11 years. The corresponding standard deviation is displayed as yellow band in Fig. 8. A linear regression fit on the smoothed model mean tropical tropopause height yields a rise of $(0.81 \pm 0.01)\,\mathrm{hPa\,decade^{-1}}$. This is in accordance with results from ECMWF Re-Analysis data for the past two decades (Wilcox et al., 2012). As indicated by Oberländer-Hayn et al. (2016) regarding the BDC, the upward shift of the tropopause affects the interpretation of vertical profile differences between future and past. An air parcel which would have already entered the stratosphere under present day conditions may be still considered tropospheric in the future. As pointed out earlier, profiles appear shifted by a fraction of distance between two pressure coordinate levels. We perform a spline fit to the averaged profiles and shift them accordingly with respect to the mean tropopause. The fit results

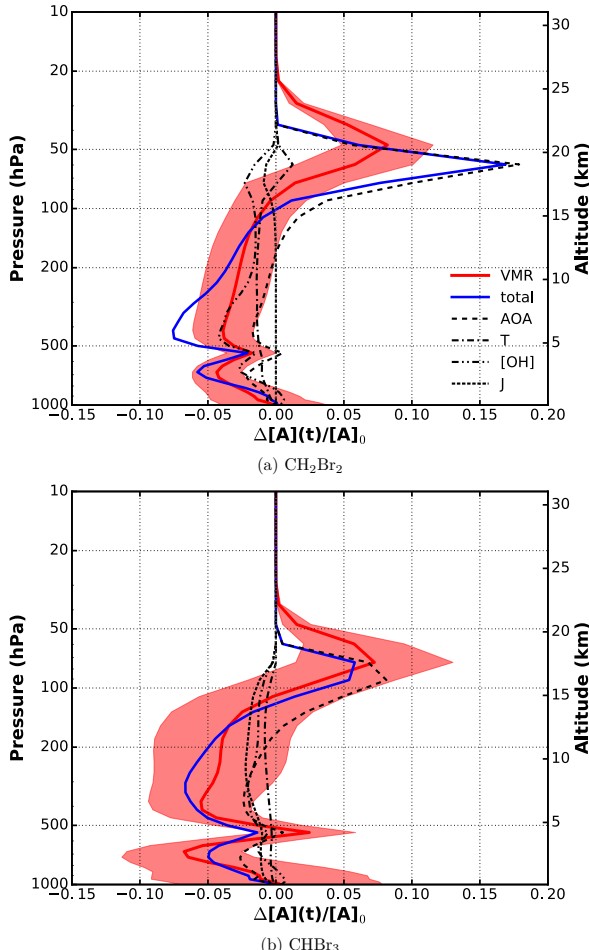

**Figure 7.** Relative difference of VSLS vertical profiles for 2000 and 2100. Major influences on lifetime have been separated. Shown are resulting profiles by varying the denoted variables, mean temperature $T$, OH concentration [OH], photolysis frequency $J$, and age of air (AOA), in Eq. (3) one by one.

have been evaluated within a valid region of $\pm 100\,\mathrm{hPa}$ around the tropopause. The results are shown in Fig. 9. Uncertainty bands have been estimated by adding/subtracting one standard deviation from the averaged VMR profiles and computing the corresponding splines. With respect to the mean tropopause, VMR differences show no increase of bromine from VSLS in the lower stratosphere, but rather a slight decrease (Fig. 9a). A small increase of inorganic bromine from VSLS ($\mathrm{Br}_y^{\mathrm{VSLS}}$) is found

5   in the tropopause region. At about $20\,\mathrm{hPa}$, $\mathrm{Br}_y^{\mathrm{VSLS}}$ is reduced by $1$–$2\,\mathrm{ppt}$ in the future compared to 1980. Overall, a reduction of bromine in the UTLS is found at the end of the 21st century. In Fig. 9b, the amount of bromine from $CH_3Br$ and Halons is shown. Except for a slight increase of $\mathrm{Br}_{\mathrm{tot}}^{\mathrm{CH_3Br+Halons}}$ in the upper stratosphere between 1980 and 2100 of about $0.7\,\mathrm{ppt}$, there is no increase of bromine from long-lived SG.

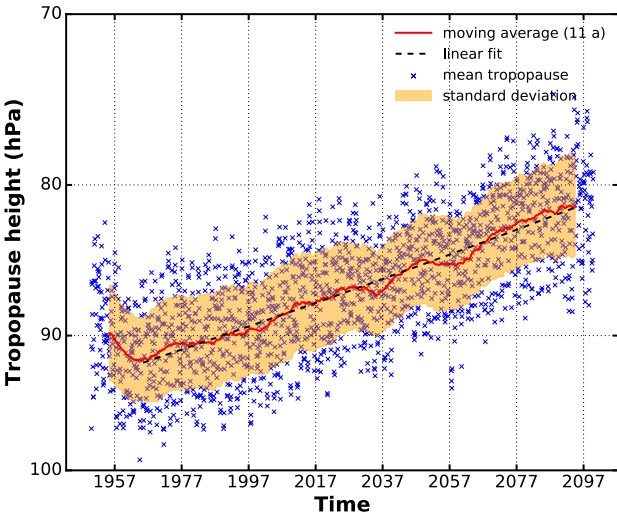

**Figure 8.** Model mean tropical tropopause from RC2-base-05 over a time span of 150 years. Tropopause data have been evaluated after the apparent spin-up of about 10 years. A rise of the mean tropical tropopause of $(0.81 \pm 0.01)\,\mathrm{hPa\,decade^{-1}}$ is found by linear regression.

To summarize, the increase of lower stratospheric VSLS in RC2-base-05 of about 5–10% is due to enhanced vertical transport in the tropics. This increase is, however, counteracted by a corresponding decrease in inorganic bromine. Everything else unchanged, an increase in tropical upwelling will therefore not change the total amount of bromine in the future stratosphere. Additionally, due to enhanced future OH concentrations in RCP6.0, the tropospheric lifetime of VSLS is reduced which leads

5  to a decrease of total bromine from VSLS. As mentioned in Section 3, whether the amount of inorganic PG in the UTLS is decreasing or not, strongly depends on the partitioning of $\mathrm{Br_y}$ and conversion of soluble HBr and HOBr into insoluble BrO through heterogeneous recycling, e.g., occurring on sea-salt aerosols or ice-crystals. In case insoluble species are favored, vertical transport would enhance the amount of PG in the UTLS. Otherwise, wet removal in the troposphere would decrease the amount of PG. This mechanism has not been explicitly tested in our model simulations. Taken an upward shift of the

10  tropopause into consideration and shifting the VMR profiles accordingly with respect to the mean tropical tropopause height, a decrease of $\mathrm{Br_{tot}^{VSLS}}$ by 0.5–2 ppt is found for a fixed $\Delta_{\mathrm{TP}}P \approx 20\,\mathrm{hPa}$.

## 5   Implications on ozone depletion

In this section, the influence of brominated very short-lived source gases on ozone depletion will be discussed. Based on RT1a and RT1b, we assess the impact of VSLS on a zonally averaged ozone distribution at the end of the 21st century. For

15  a thorough discussion of future trends, the 25 year data set is too short. However, from our long-term simulations (SC_free, SC_nudged, RC2-base-05), long-term influence of emission perturbations on ozone is not assessable. As described in Section 2, SC_free does not include interactive ozone chemistry, whereas RC2-base-05 incorporates prescribed fluxes based on scenario

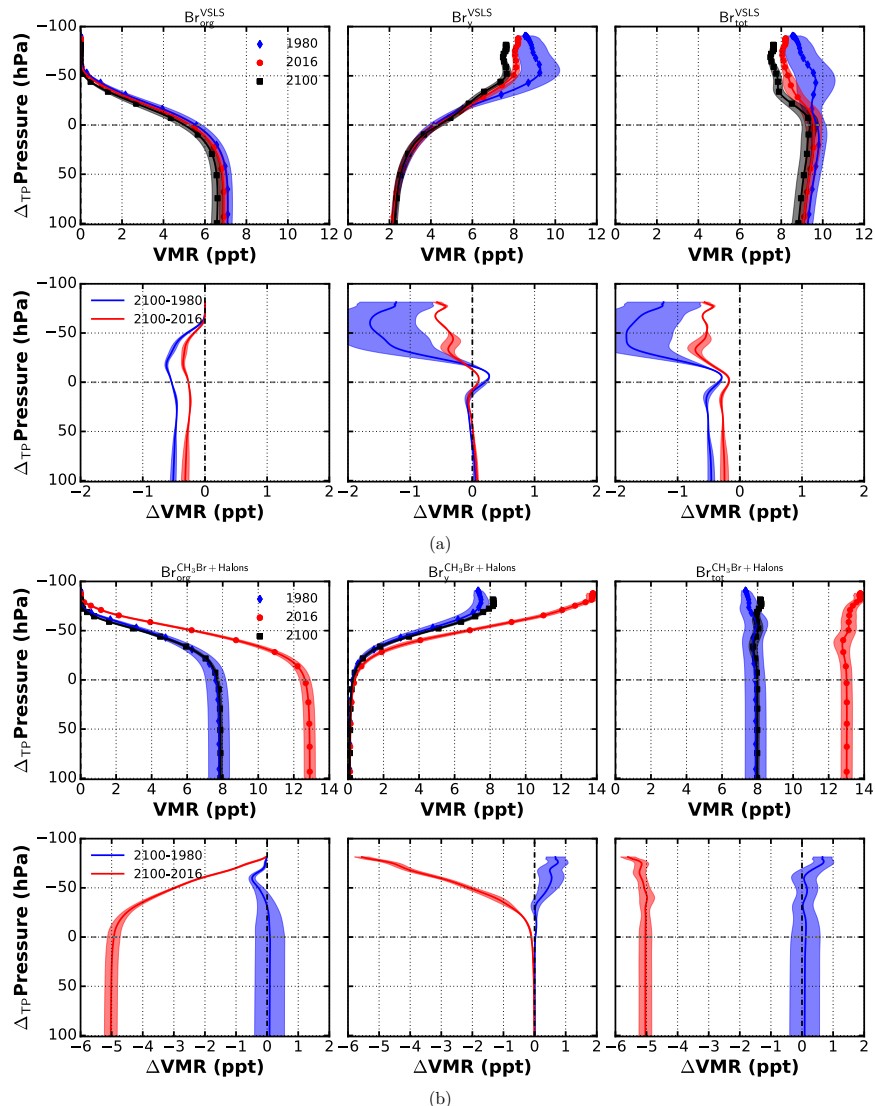

**Figure 9.** Spline fitted vertical profiles of brominated substances divided into SG ($Br_{org}$), PG ($Br_y$), and SG + PG ($Br_{tot}$) in the tropics (20°N–20°S) with respect to the mean tropical tropopause. Data from ESCiMo RC2-base-05 simulation (Jöckel et al., 2016). Absolute values of VMR in upper panel, difference $\Delta$VMR with respect to 2100 values in lower panel. (a) Bromine from VSLS; (b) Bromine from $CH_3Br$ and Halons.

five by Warwick et al. (2006). Furthermore, results from RC2-base-05 cannot be compared to RT1a/RT1b directly, because of significant differences in ozone distribution and amount between the differing vertical resolutions (L47MA and L90MA) of the model. This issue has been already reported by Jöckel et al. (2016).

Zonally averaged data of total column ozone have been smoothed using a moving average algorithm with box window size

of 11 years (Fig. 10). In general, ozone trends at the end of the century are roughly the same for both resolutions. The actual amount of ozone differs, with L90 showing more ozone except for the northern hemisphere polar region and mid-latitudes. In the Arctic RT1a/RT1b even indicate, in contrast to RC2-base-05, slightly decreasing total column ozone. In case of RT1a/RT1b, this might be partially caused by interactive aerosol and accordingly added oceanic COS source. This bias in total column ozone

between the vertical resolutions is larger than the difference between RT1a and RT1b.

For estimating the effect of brominated VSLS on ozone depletion, the difference in zonally averaged ozone of RT1a and RT1b has been computed. A period of 20 years (2080–2100) has been used accounting for an estimated model spin-up of five years in the beginning. In Fig. 11, the relative difference $((RT1a - RT1b)/RT1a \cdot 100)$ is shown as contour plot. Dashed lines indicate a decrease of ozone in the simulation with VSLS turned on (RT1a) compared to the one with VSLS turned off (RT1b), while

solid lines indicate an increase. Significance has been estimated as divergence from zero in units of standard error of mean. It is indicated by shades of blue. VSLS cause a tropospheric ozone reduction in the order of 1–2%, mainly at high latitudes. The UTLS region in the tropics is most affected, there VSLS cause a decrease of ozone of about 3%. The decrease of ozone in the high latitude troposphere and tropical UTLS is rendered significant. Increasing amounts of ozone ($\sim$1%) are found in the Antarctic middle and upper stratosphere, but these are mainly not significant. This increase in ozone abundance may be due to

dynamical feedback (Braesicke et al., 2013).

While VSLS have a large impact on Antarctic ozone depletion during the Ozone Hole period, i.e. during times with high stratospheric chlorine loading from about the late 1970s to the second half of the 21st century (e.g. Fernandez et al. (2017); Oman et al. (2016); Sinnhuber and Meul (2015); Yang et al. (2014)), we find that by the end of the 21st century under low chlorine loading VSLS have less impact on total Antarctic stratospheric ozone depletion, although their importance relative

to the total stratospheric halogene load is increasing (about 40% in accordance to Fernandez et al. (2017)). Assuming an adherence to the Montreal protocol, stratospheric volume mixing ratios of $Cl_y$ will decrease exponentially in the course of the 21st century from its peak values in 2000. From the Global Ozone Research and Monitoring Project (2014, Chap.2, Fig. 2–21), a decline of $Cl_y$ loading at $1\,hPa$ of about $2\,ppbv$ between 2000 and 2080 can be deduced. In accordance, we find a reduction of zonally averaged stratospheric $Cl_y$ at $1\,hPa$ of $2.1\,ppb$ by the end of the 21st century compared to year 2000 in

RC2-base-05. Since RT1a/RT1b have identical chlorine load and do not include present day, we cannot assess the chlorine moderation effect from these two simulations. Sinnhuber and Meul (2015) have shown a reduction of ozone due to VSLS in the TTL region in the order of 6% during the period 1970–1982 with significantly less stratospheric chlorine compared to the later period 1983–2005 (7%), while Hossaini et al. (Supplement 2015, Fig. S3) show a change of total ozone column due to VSLS (on/off scenario) of the order of 1–3% in pre-industrial conditions. Given that the VSLS emission scenario five

of Warwick et al. (2006) used in RC2-base-05 has about twice the amount of VSLS compared to RT1a and chlorine abundance in the stratosphere will drop to 1970 values by the end of the 21st century (see IPCC - Intergovernmental Panel on Climate Change, 2013, Chap. 12), our results are in good agreement with these previous studies. In more detail, Yang et al. (2014) investigated the combined influence of brominated VSLS and chlorine on ozone. For two stratospheric $Cl_y$ loadings ($3\,ppb$, $0.8\,ppb$), which correspond to 2000 and 2100, they have varied the amount of VSLS. Yang et al. (2014) have shown the more

chlorine in the stratosphere the stronger ozone is affected by an increase of bromine VMR from VSLS. In concert with our

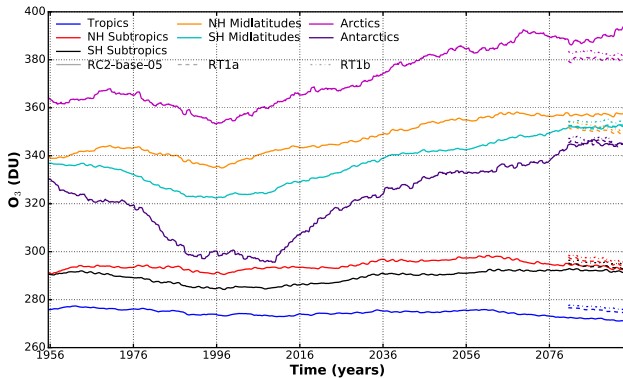

**Figure 10.** Zonal mean ozone trend from RC2-base-05 (1950–2100) and RT1a/RT1b (2080–2100). Smoothed with moving average box window 11 years.

results, although by doubling the initial amount of VSLS on a varying bromine background from anthropogenic sources, they have found a significant decrease of ozone in the tropical UTLS and polar region in the order of 2–4% and slight, insignificant increases in the antarctic mid-stratosphere (Yang et al., 2014, Fig. 1e).

## 6 Conclusions

We have investigated long-term changes in emission and transport of brominated VSLS under a changing climate (RCP6.0). Under the implicit assumption of constant concentrations of VSLS in the ocean waters, over a time-span of 120 years, we have found an enhancement of zonally averaged fluxes of $CH_2Br_2$ and $CHBr_3$ in the order of 10% between present day and the end of the 21st century. A strong increase of flux (up to 55% in $CH_2Br_2$ and 25% in $CHBr_3$) has been found in the northern hemisphere polar region. There, the retreat of sea ice is playing a key role. Exposing almost the entire polar ocean in August–September by the end of the 21st century, sea ice does not longer act as a lid to ocean–atmosphere fluxes of VSLS. Sea ice itself has not been considered a source of VSLS in our simulations. Subsequently, an increase of organic bromine in the UTLS is found of the same order of magnitude (8–10%).

Ocean–atmosphere fluxes are sensitive to the abundance of VSLS in the atmosphere as well as on wind speed. An increased dissociation of VSLS in the lowermost troposphere, e.g., due to increasing OH concentrations in the RCP6.0 scenario, reduces the atmospheric concentration and therefore increases the flux from the ocean to the atmosphere without necessarily increasing the actual amount which is transported to the stratosphere. The total amount of bromine from VSLS transported through the UTLS strongly depends on the washout of inorganic PG ($Br_y^{VSLS}$) and hence on the partitioning and heterogeneous reactions converting $Br_y^{VSLS}$ between soluble, e.g., HBr, HOBr, and insoluble, e.g., BrO, species. But these mechanisms have not been subject to our study.

For prescribed, constant VSLS fluxes, an increase of lower stratospheric VSLS of about 5–10% is due to enhanced vertical transport in the tropics. This increase is counteracted by a corresponding decrease in inorganic bromine. Everything else un-

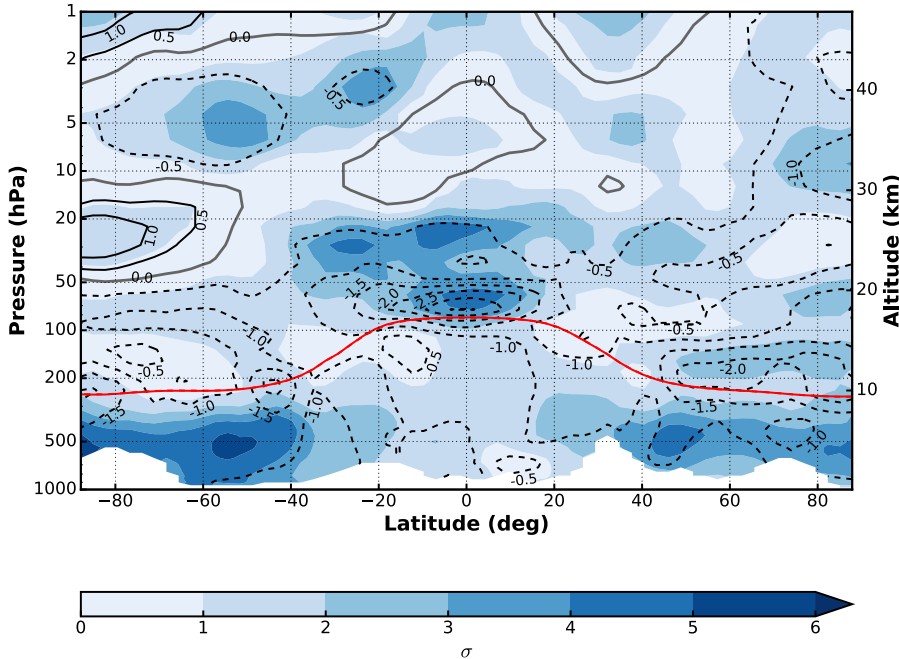

**Figure 11.** Ozone reduction due to VSLS. Medium-term full-chemistry future simulation with and without VSLS. Zonal mean profile of decadal mean difference in percent $((\mathrm{RT1a} - \mathrm{RT1b})/\mathrm{RT1a} \cdot 100)$. Dashed lines indicate a decrease of ozone in the simulation with VSLS turned on (RT1a) compared to the one with VSLS turned off (RT1b). The average tropopause is shown as red line. Significance is indicated by blue shaded areas. The significance is estimated as manifold of difference from zero in units of standard error of mean difference.

changed, an increase in tropical upwelling will not change the total amount of bromine in the future stratosphere. Additionally, due to enhanced future OH concentrations in the RCP6.0 emission scenario, the tropospheric lifetime of VSLS is reduced, which leads to a decrease of total bromine from VSLS. Furthermore we have diagnosed a decrease of $\mathrm{Br}_{\mathrm{tot}}^{\mathrm{VSLS}}$ by 0.5–2 ppt for a fixed pressure level with respect to the mean tropical tropopause $\Delta_{\mathrm{TP}}P \approx 20\,\mathrm{hPa}$, if the upward shift of the mean tropical

5  tropopause of $(0.81)\,\mathrm{hPa\,decade^{-1}}$ is taken into consideration.

The impact of enhanced fluxes of brominated VSLS on future ozone abundance has been evaluated by comparing two experiments of which one has no VSLS emission and the other interactively computed fluxes from constant ocean concentrations of VSLS. We have found a significant reduction of ozone in the tropical UTLS of about 3%. In the troposphere the largest significant decrease of ozone amounts to 1–2%. Thus, bromine from VSLS may not act as a major source to future stratospheric

10  ozone depletion. While interactive emissions from constant ocean concentrations have been taken into consideration, the actual climate change inflicted change in the production of VSLS by macroalgae in the ocean remains an open question. Whether the found increase of ocean–atmosphere fluxes of VSLS and a future decrease of VSLS in the troposphere will cancel out or overcompensate would need further simulation studies.

## 7 Code availability

The Modular Earth Submodel System (MESSy) is continuously further developed and applied by a consortium of institutions. The usage of MESSy and access to the source code is licensed to all affiliates of institutions, which are members of the MESSy Consortium. Institutions can become a member of the MESSy Consortium by signing the MESSy Memorandum of Understanding. More information can be found on the MESSy Consortium Web-site (http://www.messy-interface.org).

## 8 Data availability

The data of the ESCiMo simulations will be made available in the Climate and Environmental Retrieval and Archive (CERA) database at the German Climate Computing Centre (DKRZ; http://cera-www.dkrz.de/WDCC/ui/Index.jsp). The corresponding digital object identifiers (doi) will be published on the MESSy consortium web-page (http://www.messy-interface.org). A subset of the data of those simulations covering consistently the requested time periods (1960–2010 for RC1, and 1960–2099 for RC2) will be submitted to the BADC database for the CCMI project.

Data from ROMIC–THREAT associated simulations (RT1a, RT1b) and simplified chemistry (SC_free, SC_nudged) will be made available on request.

*Author contributions.* S. Falk performed most of the analyses and wrote the paper. B.-M. Sinnhuber conceived this study and provided advice through discussion of the analysis and results. G. Krysztofiak developed and performed the simplified chemistry simulations as well as part of the corresponding data analysis in Sec. 3. S. T. Lennatz provided advise on the ocean–atmosphere gas exchange. P. Jöckel provided advice as project leader of the ESCiMo consortial project and coordinator of overall EMAC model development; preparation of the ESCiMo model setups and realization of the ESCiMo simulations of ESCiMo consortium, with VSLS boundary conditions and implementation of the online Br budget diagnostics for EMAC prepared by P. Graf. All coauthors contributed to the discussion of the results.

*Competing interests.* The authors declare that they have no conflict of interest.

*Acknowledgements.* Parts of this work were supported by the Deutsche Forschungsgemeinschaft (DFG) through the research unit 'SHARP' (SI1044/1-2), the German Bundesministerium für Bildung und Forschung (BMBF) through the project 'ROMIC-THREAT' (01GL1217B), the European Union through the Horizon 2020 project 'GAIA-CLIM', and by the Helmholtz Association through its research program 'ATMO'.

The CNRM data were produced in the framework of the CCMI project, with support of Météo–France. We particularly acknowledge the support of M. Michou and D. Saint–Martin and of the entire team in charge of the CNRM/CERFACS climate model.

NOAA Optimum Interpolation (OI) V2 fields were provided by the National Centers for Environmental Prediction/National Weather Service/NOAA/U.S. Department of Commerce, and National Climatic Data Center/NESDIS/NOAA/U.S. Department of Commerce research Data Archive at the National Center for Atmospheric Research, Computational and Information Systems Laboratory. http://rda.ucar.edu/

datasets/ds277.0/. Accessed 06 January 2016).

The ESCiMo (Earth System Chemistry integrated Modelling) model simulations have been performed at the German Climate Computing Centre (DKRZ) through support from the BMBF. DKRZ and its scientific steering committee are gratefully acknowledged for providing the HPC and data archiving resources for this consortial project.

EMAC simulations RT1a/1b have been performed at Steinbuch Center for Computing at KIT. Thanks to Stefan Versick and Oliver Kirner (KIT SimLab Climate and Environment) for their technical support.

Special thanks to C. Brühl (MPI-Mainz) for his help in implementing additional sulfur reactions and usage of `gmxe` in context of the ROMIC–THREAT simulations.

S. Lennartz likes to thank B. Quack, C. Marandino, S. Tegtmeier (all Helmholtz-Centre for Ocean Research Kiel), and K. Krüger (University of Oslo) for their support.

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
