# Peer review of "Brominated VSLS and their influence on ozone under a changing climate"

_Atmospheric Chemistry and Physics, 2017_

## Referee Comment (RC1) · Anonymous Referee #2 · 31 Mar 2017

**General comments:**

Falk and co-authors used EMAC model to investigate brominated VSLS and their influence on ozone under a changing climate. The IPCC scenario RCP6.0 was chosen to represent the future climate. The emission of VSLS for the 21[st] century was calculated based on fixed VSLS concentrations in the ocean surface layer. Through carefully designed experiments, they attempted to separate the individual contribution from each of the key factors (T, dynamical transport, [OH] chemistry, tropopause, AOA, etc) that may affect bromine supply from VSLS in both the troposphere and the stratosphere. This manuscript highlights some interesting insights of how these factors may affect VSLS as a source of bromine in a changing climate and the corresponding ozone change due to the VSLS is well in line with previous simulations. Although there are a few 'flaws', e.g. in model setup (see below), the manuscript is well written and fluent to read, and the model conclusions sounds robust. As a conclusion, I support publication in ACP, but some clarifications are needed (see below).

**Specific comments:**

The assumption of constant oceanic concentrations for VSLS under RCP6.5 scenario is obviously a source of bias. It is true that we have very limited information about the possible responses of marine ecosystem to a warming climate. However, the manuscript could benefit from inclusion of discussions about how sensitive the VSLS emission could be to water concentrations (as well as to SST or wind). What are the responses of stratospheric ozone to the emission perturbations, in a linear or non-linear way? Could the authors derive something from their long-term integrations (without making any further experiment)? If not, then what the literatures say on the same question?

My second concern is about chlorine (Cly) effect. In the introduction section, the authors have clearly addressed the importance of mixed halogen reactions (Br-Cl) to the stratospheric ozone. However, this topic was not explicitly touched or discussed in the main text. The Cly difference from its peak to that at the end of the 21[st] century could be a few ppbvs. Is it not enough to make any difference in the Br-Cl-ozone system? Is the Cl-effect in the model result un-detectable? If so, why? Please supply the Cly curves in the revision.

P1 Line 1: Traditionally, the abbreviation 'VSLS' represents 'Very short-lived substances', not 'very short-lived source'. How about change the first sentence to 'Source gases of very short-lived substances (VSLS) contribute significantly to... '. A same problem is also spotted in P2 Line 4.

P2 Line 6-7: the '($C_w$)' following 'atmosphere' should be '($C_{air}$)'. Since both $C_w$ and $C_{air}$ are not used here (they are later re-defined on Page 5 in equation 1), thus they can be removed from the text here.

P5 Line 12-15: the equation 1 is not fully described. For example, $K_w$ and $K_{air}$ are not clearly defined; 'temperature T' in line 15 should be 'air temperature $T_{air}$'

---

## Referee Comment (RC2) · Anonymous Referee #1 · 18 Jul 2017

**Review of Brominated VSLS and their influence on ozone under a changing climate by Falk et al., ACPD, 2017**

The paper presents an interesting modeling approach to evaluate how the future changes on temperature, dynamics and composition of the atmosphere will impact on the ocean-atmospheric source strength of very short-lived (VSL) source gases, as well as on the evolution of the tropopause height and stratospheric ozone. Based on a wide set of model experiments, an evaluation of different factors such as photolysis, OH reactivity, Age of Air (AOA) and temperature affecting the degradation of the most abundant VSL species ($CHBr_3$ and $CH_2Br_2$) are analyzed. The main results presented indicate that even when the VSLS ocean-atmospheric flux will increase ~10% between present time and 2100, the total amount of bromine from VSLS in the stratosphere will decrease during the 21[st] century.

I found the paper interesting and well diagramed, presenting results in a clear and complete format. But there are two major concerns that might be affecting the conclusions obtained from this work that must be clarified before final publication. One of them is regarding the relation of present EMAC modelling results with other published works in the literature with equivalent assumptions, and how the current model configuration could be affecting the interpretation of the VSLS flux evolution. The other main issue is regarding the tropospheric treatment of the inorganic product gas (PG) fraction and how additional factors/processes that are not considered in their analysis might be affecting the evolution of inorganic bromine injection. I believe the authors should be able to answer both of the main issues without the need of any further sensitivity simulation or changes in their model setup, so the paper can be accepted once those changes are included without any further review. At the end, several minor comments are given a) asking for relevant configuration details that are omitted, b) highlighting some sentences in the text that are not clear and should be modified and c) indicating where further discussions are required.

**Major Comments:**

1. There is a notorious omission to a strongly related paper from Ziska et al., (2017), which used exactly the same methodology to address the future evolution of the ocean-atmosphere flux of VSL through the 21[st] century. Even when the current study goes beyond the above-mentioned work by addressing the atmospheric factors controlling the VSL stratospheric injection and impact on ozone, a comparison and description of the similarities and differences regarding the Ziska et al. 2017 paper should be given.
*(Ziska, F., Quack, B., Tegtmeier, S. et al. J Atmos Chem (2017). Future emissions of marine halogenated very-short lived substances under climate change. 74: 245. doi:10.1007/s10874-016-9355-3)*
In particular:
   - Ziska et al. considered the RCP 2.6 and RCP 8.5 scenarios, and determined a linear response only for the 1979-2005 period, but not when projecting into the future. Additionally, their RCP 2.6 increase of brominated emissions (9%) is of the same magnitude as the one found here for RCP 6.0 (10%).

- P3,L3: "In these simulations, OH concentrations have been set to zero in the lower troposphere (700–1000 hPa) to reduce the variability of ground level volume mixing ratio (VMR) of VSLS". Thus, degradation of $CHBr_3$ and mostly $CH_2Br_2$ in the MBL and Lower troposphere is not well represented and might affect their tropospheric concentration ($C_{air}$). But eq. 1, which infers the future evolution of VSLS emissions, depends on the $C_{air}$ concentration, which would be larger than if OH values would have not been forced to zero in the MBL. Could this assumed configuration be related to the similar % between RCP 2.6 from Ziska et al., 2017 and present work?

2. Consideration of heterogeneous recycling of $Br_y^{VSL}$ in the UT and TTL might be of major importance in current study, and there is not a single mention of it neither in the model configuration nor in the results analysis. Many modelling studies, including some performed by the same group (Aschmann et al., 2009), other cited in the text (Liang et al., 2014) and many others not even mentioned in the manuscript (Parrella et al., 2012; Fernandez et al., 2014; Wang et al., 2015; Schmidt et al., 2016) highlight the importance of considering heterogeneous recycling occurring on ice-crystals and sea-salt aerosols as they can increase the lifetime against wet-removal or represent an additional source of bromine to the troposphere, respectively. Indeed, Fig. 5 and Fig 9. clearly shows that $Br_y^{VSLS}$ is the dominant fraction controlling the total stratospheric $Br_{Tot}^{VSLS}$ change between present time and 2100, which highlight the importance of properly representing inorganic product gases chemistry in present study.

   - Authors should decide whether performing additional simulations including and neglecting the heterogeneous recycling reactions is necessary or not. But if they instead want to focus on VSL source gases, at least the paper must mention how the uncertainties of heterogeneous recycling processes could be affecting their overall stratospheric results. A very rapid analysis could indicate that because of the increased tropospheric degradation, there is a larger $Br_y$ fraction that is effectively washed out from the troposphere and never reaches the stratosphere. Thus, reducing the overall PG stratospheric injection. If that is the case, then it should be explicitly mentioned in the text, supported with more details about the deposition efficiencies of $Br_y$ species.

   - Abstract, P1,L7: "A decrease in the tropospheric mixing ratios of VSLS and an increase in the lower stratosphere are attributed to changes in atmospheric chemistry and transport. Our model simulations reveal that, in line with the reduction in the troposphere, the total amount of bromine from VSLS in the stratosphere will decrease during the 21st century". I found a contradiction between these two sentences included in the abstract, which is not clarified nor consistent with the final sentence in Section. 4.2

   - Section 4.2, P13,L15: "To summarize, the main reason to the apparent increase of bromine from VSLS above 100 hPa in RC2-base-05 of about 5–10% is the increase in vertical transport in the tropics. Although bromine loading from VSLS above 100 hPa is increased at the end of the 21st century, the stratospheric abundance of bromine from VSLS is not increasing but decreasing by about 1–2 ppt, if an upward shift of the tropopause is taken into consideration." This summarizing result seems not to be consistent with the rest of the text nor what is shown in the figures. The

- only 1-2 ppt reduction occurs in Bry PG (Fig. 9a, center), which as I have mentioned above, might not be well represented in the modeling study and might be altering the interpretation of the results. Additionally, the 1-2 ppt difference appears for $\Delta P > 20$ hPa respect the tropopause, so changes in the tropopause height could be affecting the VSLS Bry levels in the UTLS, but not in the stratospheric over-world.

- Forcing OH to be zero in the LT also reduces the total amount of inorganic bromine product gas species being available for heterogeneous recycling. This could reduce the additional source from sea-salt dehalogenation (if considered). How this forced OH assumption could be affecting the treatment of the PG being released from aerosols?

- P6,L16: " … and therefore increase the flux from the ocean to the atmosphere without increasing the actual amount which is transported to the stratosphere." This is only the case for VSLS source gases, but if PG are not washed out right away, they could also be transported to the stratosphere.

- P8,L10: "PG from VSLS ($Br_y^{VSLS}$), which have been traced within the simulation, are decreasing in the stratosphere in the future. For 2016, this decline is compatible with a decreasing amount of VSLS in the troposphere. A slight excess of $Br_y^{VSLS}$ compared to 2016 and 2100 is found for 1980 in the stratosphere." Please considering rephrasing this sentence, or at least relate it to the impact heterogeneous recycling might have on PG.

- P9,L11: It is important to note, that although there is an apparent increase of $Br_{org}^{VSLS}$ of 0.5 ppt in the stratosphere assuming constant ocean–atmosphere fluxes, the overall amount of bromine in the stratosphere due to VSLS is decreasing in the future." Why apparent? Should you end the sentence by adding "… when the time varying fluxes are considered". Could you please explain and relate it to the Bry PG heterogeneous treatment?

- P9,Fig.4: The Figure panel indicates 1980 instead of 1990. The caption says "… an increase of roughly 10% in $Br_y$ from VSLS in the stratosphere is found, while the increase in $Br_{org}$ amounts 8%". First, could you show the tropopause height for each year in the Figure. Second, the 8% $Br_{org}$ increase is at the tropopause?, Third, this sentence seems to contradict P1,L8 in the abstract that total amount of bromine from VSLS in the stratosphere will decrease during the 21st century.

- There is a contradictory message between P10,L1 ", the overall amount of bromine in the stratosphere due to VSLS is decreasing in the future" and P10,L9 "The increase of VSLS in the stratosphere in the future can be attributed to". Are VSLS increasing or decreasing in the future stratosphere??

- Conclusions, P18,L21: "Due to the rise of the tropical tropopause by 0.81 hPa decade$^{-1}$, air which at present is considered stratospheric will be still tropospheric in the future. If taken into account by shifting VSLS VMR profiles with respect to the mean model tropopause height, the total amount of bromine in the tropical UTLS is decreasing by roughly 2 ppt. Overall, the amount of bromine in the UTLS is decreasing in this future scenario." The changes in VSLS bromine are only affecting the UTLS or the overall stratosphere? Please make it clear and consistent with the abstract and main text in line of all abovementioned concerns.

3. Also, comparison with the results of a recent study (Fernandez et al., 2017) that estimated the effect of biogenic VSLBr species in the evolution of the Antarctic ozone hole during the 21$^{st}$ century should be made.
(*Fernandez et al., Impact of biogenic very short-lived bromine on the Antarctic ozone hole during the 21st century, Atmos. Chem. Phys., 17, 1673-1688, 2017*)

**Minor Comments:**

P2,L20: "… how transport and *tropospheric* chemistry influence the stratospheric bromine abundance" … "and how *stratospheric* ozone will be affected …". I suggest adding the italic words to make the text clear.

P2,L27: Simplified and full bromine chemistry: What are the main differences between both chemical mechanism, and how the simplified treatment could be affecting the Bry production, recycling and removal. Additionally: $Br_2$ is considered in the chemical mechanism? Because it explicitly appears in P2,L12 but it doesn't in P3,L9.

P3,L7: What do you mean by "… commuted into Bry"

P3,L19: RT1a and RT1b both include online computation of aerosol formation: What types of aerosols: tropospheric or stratospheric. What type of interaction is included in the model regarding VSLS species and the aerosol module?

P4,L1: Could you briefly mention the main differences between the Wanninkhof (1992) and Nightingale et al. (2000) parameterizations of $K_w$. How these differences can be affecting the VSLS ocean-atmosphere flux?

P4,L11: "cloud coupling had not been activated". Is this of relevance only for the radiative transfer scheme? Can it affect the model wet-removal computation?

P4,L19: How many vertical levels does the model include, and how many of them belong to the troposphere and how many to the stratosphere?

P5,L20: what do you mean by "Relative to the absolute zonal fluxes"?. Is it the global mean?

P5,L30: "Distinct maxima in the seasonal cycles … and minima occurring in late winter". Please rephrase.

P5,L31: "In case of $CH_2Br_2$, even negative emissions are found during winter at high-latitudes on the northern hemisphere.". First, you could explicitly indicate that negative emissions represent a net sink of atmospheric VSLS. Second, how the forcing of LT OH to zero could be affecting this negative flux?

P6, Table 2 caption: "Average absolute flux for year 2000 …"

P6,L11: I suggest indicating also the absolute increase in VSLS surface VMR.

P5,L15 and P6,L14: The dependence of the emission flux on wind speed is not explicitly mentioned in Eq. 1.

P6,L17: "Much stronger fluxes (1.3–1.5 times) have been found in the former simulation in comparison to the latter." First, could you rephrase to make it clear which is the former and which the latter. Second, could the flux difference be due to the different OH zeroing treatment between experiments within the MBL and LT?

P8,Fig 3: How did you set the $C_w$ from the Ziska et al., 2013 paper for regions in the Artic that where covered by sea-ice at present time but are not longer covered in the future?

P12,L18 and elsewhere in the text: is it *ansatz* an accepted English word?

P13,Fig.6: Have you thought about including the $CH_2Br_2$ and $CHBr_3$ photolysis rate vertical profile in a second panel?

P13,L11: "At about 20 hPa" … Do you mean a 20hPa difference from the mean tropopause? Regarding Fig. 9, and considering P8,L1 "There is an upward shift of the tropopause height of about 8 hPa between present day and future". Why do you show such a large shift in pressure respect to the mean tropopause (± 100 hPa) if the difference in the tropopause pressure is smaller than 10 hPa? I would expect the changes in the tropopause height to affect only the UTLS, and the partitioning between SG and PG, but not the overall total bromine abundances in the middle and upper stratosphere.

P15,L14: If the authors are willing to address the impact of VSLS in the future evolution of Antarctic ozone, they should at least compare their results respect to Oman et al., 2016 and Fernandez et al., 2017.

P18,L29: "… and aerosol formation have been taken into consideration". While a full aerosol treatment has been considered for some of the simulations, the sentence gives the impression that an aerosol formation module for VSLS has been considered in this work. I suggest rephrasing to avoid misleading interpretations.

---

## Author Response (AR1)

**Authors' response**

acp-2017-34-RC1-supplement (31 March 2017)

The authors thank the anonymous referee #2 for his constructive comments regarding our paper. We will answer his/her specific questions in detail in the following. Accordingly, we will provide comprehensive discussions and additional numbers in a revised version of the paper where further clarifications are needed.

– Specific comments:

- **Sensitivity of VSLS emissions wrt. water concentrations/wind/SST:** No simulation study has been conducted in the regard of changing water concentrations of VSLS *and* a change in climate. Therefore no assessment can be made. At present, global mean $CH_2Br_2/CHBr_3$ concentrations in ocean water are close to equilibrium with atmospheric concentrations which is causing a high sensitivity of ocean–atmosphere fluxes to changes in the atmosphere (Lennartz et al., 2015). There are different potential future scenarios. The current scenario could be interpreted as an increase in production in response to increased fluxes to compensate a detrainment. Increased fluxes and decreasing production at the same time may lead to decreasing concentrations and in turn decreasing fluxes. Regarding a dependence of VSLS fluxes wrt. SSTs and wind, our data show a strong correlation between VSLS fluxes and former ($\Phi_{VSLS}(SST) \propto SST$), but a much weaker (or even anti-) correlation to latter.

- **Emission perturbation's influence on ozone:** Unfortunately, our long-term simulations do not allow for an assessment of ozone changes induced through VSLS emission perturbations. SC_free does not include interactive ozone chemistry, whereas RC2-base-05 incorporates prescribed fluxes based on scenario five by Warwick et al. (2006).

- **Chlorine moderation of bromine influence on ozone:** Future projections of stratospheric chlorine loading are shown, e.g., in IPCC - Intergovernmental Panel on Climate Change (2013, Chap. 12) or Global Ozone Research and Monitoring Project (2014, Chap.2). Stratospheric $Cl_y$ peaks in 2000. Assuming an adherence to the Montreal protocol, stratospheric volume mixing ratios of $Cl_y$ will decay exponentially in the course of the 21st century. From Global Ozone Research and Monitoring Project (2014, Chap.2, Fig. 2–21), a decrease of $Cl_y$ loading at 1 hPa of about 2 ppbv between 2000 and 2080 can be deduced. We will provide numbers of peak and end of 21st century values for the $Cl_y$ load in the stratosphere in accordance to our simulations. We acknowledge, the combined effect of chlorine and bromine is not negligible. However, it can not be assessed with our simulations. RC2-base-05 is not accompanied by a sensitivity study with no VSLS emission. RT1a/RT1b have the same chlorine load and do not include present day. The moderation of differing chlorine loading on bromine influence on ozone has been studied in detail by Yang et al. (2014) and Sinnhuber and Meul (2015). Yang et al. (2014) show for two stratospheric $Cl_y$ loadings (3 ppb, 0.8 ppb) corresponding to 2000 and 2100 values and differing VSLS contribution to stratospheric bromine loading, the more chlorine the stronger ozone is affected by increasing bromine mixing ratios from VSLS. Between the two $Cl_y$ scenarios a $\Delta O_3$ of about 0.5–0.6 ppmv (80° S) and 0.3–0.4 ppmv (80° N) has been found.

  **P1 L1:** The sentence has been changed in accordance to the comment: *Very short-lived substances (VSLS) contribute as source gases [...]*

  **P2 L4:** Accordingly, the acronym definition has been changed and moved to the proper sentence: *Minor brominated very short-lived substances (VSLS) include [...]. The tropospheric lifetime of these gases lies between several days to weeks.*

  **P2 L6–7:** The definitions at this point have been removed since they are, indeed, redefined later on.

  **P5 L12–15:** In accordance to the comments a specification of temperature and transfer velocities has been added: *The transfer velocity depends largely on temperature $T_{air}$ and surface wind speed which is taken into account by distinguishing between water- and air-side transport velocities ($k_w$, $k_{air}$).*

The authors thank the anonymous referee #1 for his/her thorough work and stimulating comments regarding our paper. Detailed response to the posed questions will be given in the following.

– Major comments:

5    1. There is a notorious omission to a strongly related paper from Ziska et al. (2017), which used exactly the same methodology to address the future evolution of the ocean-atmosphere flux of VSL through the 21st century. Even when the current study goes beyond the above-mentioned work by addressing the atmospheric factors controlling the VSL stratospheric injection and impact on ozone, a comparison and description of the similarities and differences regarding the Ziska et al. (2017) paper should be given. We have include a discussion of the Ziska et al.
10    (2017) results and how they relate to our work. Ziska et al. (2017) was published online on 29 December 2016, only a few days before our submission.

• Ziska et al. considered the RCP 2.6 and RCP 8.5 scenarios, and determined a linear response only for the 1979–2005 period, but not when projecting into the future. Additionally, their RCP 2.6 increase of brominated emissions (9%) is of the same magnitude as the one found here for RCP 6.0 (10%). There is a fundamental
15    difference between the Ziska approach an ours: Ziska et al. (2017) diagnosed the flux from parameters such as SST and wind speed for a fixed concentration gradient. In our work, we compute the flux, consistently, i.e. based on changing atmospheric concentrations under the assumption of a prescribed ocean water concentration. From a theoretical point of view it is clear that we will compute a smaller increase in flux with our approach.

20    • P3 L3: "In these simulations, OH concentrations have been set to zero in the lower troposphere (700—1000 hPa) to reduce the variability of ground level volume mixing ratio (VMR) of VSLS". Thus, degradation of $CHBr_3$ and mostly $CH_2Br_2$ in the MBL and Lower troposphere is not well represented and might affect their tropospheric concentration ($C_{air}$). But eq. 1, which infers the future evolution of VSLS emissions, depends on the $C_{air}$ concentration, which would be larger than if OH values would have not been forced to zero in the
25    MBL. Could this assumed configuration be related to the similar % between RCP 2.6 from Ziska et al. (2017) and present work? First: OH was set to zero in the MBL only in the simulations SC_free and SC_nudged, not in any of the others. It is true that we overestimate the chemical lifetime of the VSLS in the MBL and that a shorter lifetime may result in a stronger flux and stronger flux increase depending on the future scenario. We will include this caveat in our revised manuscript: *Only in these simulations with simplified chemistry,*
30    OH *concentrations have been set to zero in the lower troposphere [...]. The chemical lifetime of VSLS in the lower troposphere is therefore overestimated. Due to the longer lifetime, VSLS are more abundant in the lower troposphere leading to a flux suppression.*

2. Consideration of heterogeneous recycling of $Br_y^{VSL}$ in the UT and TTL might be of major importance in current study, and there is not a single mention of it neither in the model configuration nor in the results analysis. Many
35    modelling studies, including some performed by the same group (Aschmann et al., 2009), other cited in the text (Liang et al., 2014) and many others not even mentioned in the manuscript (Parrella et al., 2012; Fernandez et al., 2014; Wang et al., 2015; Schmidt et al., 2016) highlight the importance of considering heterogeneous recycling occurring on ice-crystals and sea-salt aerosols as they can increase the lifetime against wet-removal or represent an additional source of bromine to the troposphere, respectively. Indeed, Fig. 5 and Fig. 9 clearly shows that $Br_y^{VSLS}$
40    is the dominant fraction controlling the total stratospheric $Br_{Tot}^{VSLS}$ change between present time and 2100, which highlight the importance of properly representing inorganic product gases chemistry in present study. There is, in fact, multiple evidence that heterogeneous reactions converting soluble HBr and HOBr into insoluble BrO may play a role. We did not explicitly test these mechanisms in our model simulations. We will include a discussion of the possible implications in our revised manuscript and are grateful for comments and suggested further references.

45    • Authors should decide whether performing additional simulations including and neglecting the heterogeneous recycling reactions is necessary or not. But if they instead want to focus on VSL source gases, at least the paper must mention how the uncertainties of heterogeneous recycling processes could be affecting their overall stratospheric results. A very rapid analysis could indicate that because of the increased tropospheric degradation, there is a larger $Br_y$ fraction that is effectively washed out from the troposphere and never reaches the stratosphere. Thus, reducing the overall PG stratospheric injection. If that is the case, then it should be explicitly mentioned in the text, supported with more details about the deposition efficiencies of $Br_y$ species. First, we would like to stress that the increase of lower stratospheric VSLS in the future due to enhanced upwelling is counteracted by a corresponding decrease in inorganic bromine. Everything else unchanged, an increase in upwelling will not change the total bromine in the stratosphere. In addition to this, we do find a future decrease in total bromine due to VSLS and part of this is due to the reduced lifetime of VSLS in the troposphere because of increased OH reactivity. We do, however, acknowledge that there are still structural uncertainties on the recycling of $Br_y$ in the troposphere, which may have an impact on the washout (and there are also structural uncertainties in the treatment of washout itself). We will address this caveat in the revised manuscript.

- Abstract, P1 L7: "A decrease in the tropospheric mixing ratios of VSLS and an increase in the lower stratosphere are attributed to changes in atmospheric chemistry and transport. Our model simulations reveal that, in line with the reduction in the troposphere, the total amount of bromine from VSLS in the stratosphere will decrease during the 21st century". I found a contradiction between these two sentences included in the abstract, which is not clarified nor consistent with the final sentence in Section. 4.2 Yes, these sentences are not fully consistent. We rewrite this to make our point clear: *Our model simulations reveal that this increase is counteracted by a corresponding reduction of inorganic bromine. Therefore the total amount of bromine from VSLS in the stratosphere will not be changed by an increase in upwelling.*

- Section 4.2, P13 L15: "To summarize, the main reason to the apparent increase of bromine from VSLS above 100 hPa in RC2-base-05 of about 5–10% is the increase in vertical transport in the tropics. Although bromine loading from VSLS above 100 hPa is increased at the end of the 21st century, the stratospheric abundance of bromine from VSLS is not increasing but decreasing by about 1–2 ppt, if an upward shift of the tropopause is taken into consideration." This summarizing result seems not to be consistent with the rest of the text nor what is shown in the figures. The only 1-2 ppt reduction occurs in $Br_y$ PG (Fig. 9a, center), which as I have mentioned above, might not be well represented in the modeling study and might be altering the interpretation of the results. Additionally, the 1-2 ppt difference appears for $\Delta P > 20$ hPa respect the tropopause, so changes in the tropopause height could be affecting the VSLS $Br_y$ levels in the UTLS, but not in the stratospheric overworld. Yes, the final sentences in Section 4.2 (P13 L15) are not fully consistent with the abstract. We rewrite this accordingly: *[...] the increase of lower stratospheric VSLS in RC2-base-05 of about 5–10% is due to enhanced vertical transport in the tropics. This increase is, however, counteracted by a corresponding decrease in inorganic bromine. Everything else unchanged, an increase in tropical upwelling will therefore not change the total amount of bromine in the future stratosphere. Additionally, due to enhanced future OH concentrations in RCP6.0, the tropospheric lifetime of VSLS is reduced which leads to a decrease of total bromine from VSLS. In this case, as mentioned in Section 3, whether the amount of inorganic PG in the UTLS is decreasing or not, strongly depends on the partitioning of $Br_y$ and conversation of soluble HBr and HOBr into insoluble BrO through heterogeneous recycling, e.g., occurring on sea-salt aerosols or ice-crystals. In case insoluble species are favored, vertical transport would enhance the amount of PG in the UTLS. Otherwise, wet removal in the troposphere would decrease the amount of PG. This mechanism has not been explicitly tested in our model simulations.*

- Forcing OH to be zero in the LT also reduces the total amount of inorganic bromine product gas species being available for heterogeneous recycling. This could reduce the additional source from sea-salt dehalogenation (if considered). How this forced OH assumption could be affecting the treatment of the PG being released from aerosols? OH is set to zero only in the SC_free and SC_nudged simulations in Section 3, not in any other simulation referred to in the rest of the paper. Since SC_free and SC_nudged do not use any real atmospheric

chemistry computation, these concerns are valid but do not apply to our analysis regarding these simulations. We will make this clearer in the revised manuscript.

- P6 L16: " ... and therefore increase the flux from the ocean to the atmosphere without increasing the actual amount which is transported to the stratosphere." This is only the case for VSLS source gases, but if PG are not washed out right away, they could also be transported to the stratosphere. This comment is valid. We will rephrase the sentence accordingly: *[...] increase the flux from the ocean to the atmosphere without necessarily increasing the actual amount of bromine which is transported to the stratosphere. The total amount of bromine from VSLS transported through the UTLS strongly depends on the washout of inorganic PG and hence on partitioning and heterogeneous reactions converting $Br_y^{VSLS}$ between soluble, e.g., HBr, HOBr, and insoluble, e.g., BrO, species (e.g. Aschmann et al., 2009; Liang et al., 2014). Since OH concentrations in the lower troposphere have been set to zero in SC_free, the resulting total ocean–atmosphere fluxes might be underestimated.*

- P8 L10: "PG from VSLS ($Br_y^{VSLS}$), which have been traced within the simulation, are decreasing in the stratosphere in the future. For 2016, this decline is compatible with a decreasing amount of VSLS in the troposphere. A slight excess of $Br_y^{VSLS}$ compared to 2016 and 2100 is found for 1980 in the stratosphere." Please considering rephrasing this sentence, or at least relate it to the impact heterogeneous recycling might have on PG. We rewrite the sentence to make our point clearer: *The amount of inorganic PG from VSLS ($Br_y^{VSLS}$) in the UTLS is decreasing by the same order of magnitude due to the enhanced upwelling in the tropics, as less SG ($Br_{org}^{VSLS}$) has been dissociated into PG ($Br_y^{VSLS}$), if air in the UTLS becomes younger in a future climate. For 2016, this decline is compatible with a decreasing amount of VSLS in the troposphere.*

- P9 L11: "It is important to note, that although there is an apparent increase of $Br_{org}^{VSLS}$ of 0.5 ppt in the stratosphere assuming constant ocean—atmosphere fluxes, the overall amount of bromine in the stratosphere due to VSLS is decreasing in the future." Why apparent? Should you end the sentence by adding "... when the time varying fluxes are considered". Could you please explain and relate it to the $Br_y$ PG heterogeneous treatment? First, we have removed the word "apparent" as it is a real increase in $Br_{org}^{VSLS}$. Second, the time varying flux is another issue and not related to this statement. Third, We will include a statement about the uncertainty of wet removal of $Br_y^{VSLS}$ and that it may be of importance. *[...] that although there is an increase of $Br_{org}^{VSLS}$ of 0.5 ppt in the stratosphere assuming constant ocean–atmosphere fluxes, the overall amount of bromine in the stratosphere due to VSLS ($Br_{tot}^{VSLS}$) might be decreasing in the future. This depends on whether PG ($Br_y^{VSLS}$) are transported alongside the VSLS into the UTLS or removed through washout in the troposphere. The model representation of underlying processes, e.g., conversion between soluble and insoluble inorganic bromine species through heterogeneous chemical reactions, are still uncertain.*

- P9 Fig.4: The Figure panel indicates 1980 instead of 1990. The caption says "... an increase of roughly 10% in $Br_y$ from VSLS in the stratosphere is found, while the increase in $Br_{org}$ amounts 8%". First, could you show the tropopause height for each year in the Figure. Second, the 8% $Br_{org}$ increase is at the tropopause?, Third, this sentence seems to contradict P1 L8 in the abstract that total amount of bromine from VSLS in the stratosphere will decrease during the 21st century. This Figure relates to the simulations with varying fluxes. We will make this clearer.

- There is a contradictory message between P10 L1 ", the overall amount of bromine in the stratosphere due to VSLS is decreasing in the future" and P10 L9 "The increase of VSLS in the stratosphere in the future can be attributed to". Are VSLS increasing or decreasing in the future stratosphere?? We make it clearer that VSLS increase, but the total amount of bromine due to VSLS decreases.

- Conclusions, P18 L21: "Due to the rise of the tropical tropopause by $0.81\,hPa\,decade^{-1}$, air which at present is considered stratospheric will be still tropospheric in the future. If taken into account by shifting VSLS VMR profiles with respect to the mean model tropopause height, the total amount of bromine in the tropical UTLS is decreasing by roughly 2 ppt. Overall, the amount of bromine in the UTLS is decreasing in this future scenario." The changes in VSLS bromine are only affecting the UTLS or the overall stratosphere? Please make it clear and consistent with the abstract and main text in line of all above mentioned concerns. Changes

in the tropopause altitude can explain local changes in the profiles at fixed altitudes. The diagnosed changes in tropopause altitude are not independent from the changes in transport discussed in the rest of the paper.

3. Also, comparison with the results of a recent study (Fernandez et al., 2017) that estimated the effect of biogenic VSLBr species in the evolution of the Antarctic ozone hole during the 21st century should be made. We have included a brief discussion on their results. But this does not affect any of our conclusions.

– Minor comments:

• P2 L20: "... how transport and *tropospheric* chemistry influence the stratospheric bromine abundance"... "and how *stratospheric* ozone will be affected ...". I suggest adding the italic words to make the text clear. We follow the suggestion and have added the words in italic for clarification.

• P2 L27: Simplified and full bromine chemistry: What are the main differences between both chemical mechanism, and how the simplified treatment could be affecting the $Br_y$ production, recycling and removal. Additionally: $Br_2$ is considered in the chemical mechanism? Because it explicitly appears in P2 L12 but it doesn't in P3 L9. *Simplified chemistry* means there is no interactive computation of chemical reactions, e.g, using the EMAC submodel MECCA (http://www.mecca.messy-interface.org/). VSLS degeneration to $Br_y$ is rather estimated based on lifetime wrt. fixed [OH]] and photolysis. This $Br_y$ is then converted to the listed species based on pre-computed partitioning from a previous one year long EMAC simulation with full chemistry. *Full chemistry* therefore means using interactive chemistry computation via MECCA. We make this clearer in the revised manuscript. For the second point, $Br_2$ is included in the partitioning. The absence in the text has been a typo.

• P3 L7: What do you mean by "commuted into $Br_y$"? *Commuted* means *converted*. For this was not clear, we will change the wording to *converted*.

• P3 L19: RT1a and RT1b both include online computation of aerosol formation: What types of aerosols: tropospheric or stratospheric. What type of interaction is included in the model regarding VSLS species and the aerosol module? Aerosol has been calculated prognostically with the GMXE submodel in the stratosphere and troposphere as descibed by Brühl et al. (2012, 2015). Heterogeneous reactions of VSLS dissociation product HOBr and HBr are included.

• P4 L1: Could you briefly mention the main differences between the Wanninkhof (1992) and Nightingale et al. (2000) parametrization of $k_w$? How these differences can be affecting the VSLS ocean–atmosphere flux? In both cases $k_w$ are quadratic in dependence of wind speed. Nightingale et al. (2000) is $k_w = 0.222\,u^2 + 0.333\,u$ and Wanninkhof (1992) is $k_w = 0.31\,u^2$. So, while Wanninkhof (1992) is strictly quadratic, Nightingale et al. (2000) has an additional linear term. The differences on the global level are <15% (Tab. 4 in Lennartz et al. (2015) when comparing Nightingale et al. (2000) and Wanninkhof and McGillis (1999)). Given that the mean global wind speed lies in a range where these parameterizations do not differ drastically, the uncertainty is minor. However, the Nightingale et al. (2000) parameterization reacts more sensitive to changes in wind speed, which introduces a further uncertainty when assessing changes over time in a changing climate. A note about the polynomial order of the parametrizations has been added in place. Further explanation has been also added: *The impact of various $k_w$ parametrizations on VSLS emission has been previously studied. The differences on the global level are <15% (Tab. 4 in Lennartz et al. (2015) when comparing Nightingale et al. (2000) and Wanninkhof and McGillis (1999)). For wind speeds exceeding $10\,\mathrm{ms}^{-1}$ the Wanninkhof (1992) $k_w$ parametrization diverges slightly stronger towards higher transfer velocities compared to the Nightingale et al. (2000) parametrization (cf. Fig. 1 in Wanninkhof and McGillis (1999) and Fig. 2 in Lennartz et al. (2015)). Regarding integrated global emissions of VSLS, both parametrizations result in similar fluxes, given that the mean global wind speed lies in a range where these parameterizations do not differ drastically. However, the Nightingale et al. (2000) parameterization reacts more sensitive to changes in wind speed, which introduces a further uncertainty when assessing changes over time in a changing climate.*

• P4 L11: "... cloud coupling had not been activated..." Is this of relevance only for the radiative transfer scheme? Can it affect the model wet-removal computation? The interactive coupling of tropospheric aerosol to clouds affects the

cloud properties (such as cloud cover, droplet radius, optical properties). Therefore the radiative transfer is affected, but also the aqueous-phase chemistry within cloud-droplets and, with that, also the wet scavenging.

- **P4 L19: How many vertical levels does the model include, and how many of them belong to the troposphere and stratosphere?** The number of vertical levels of the model depends on the resolution, e.g. T42L90MA, of which L90MA is an abbreviation of 90 levels, top-level in the middle atmosphere. The number of levels which belong to the stratosphere can only be stated approximately, since hybrid-pressure vertical coordinates are used in the model and the height of the tropopause varies in space and time. We have added an explanation and estimated number of levels above $100\,\mathrm{hPa}$ in the manuscript: *[...] with a top level at 0.01 $\mathrm{hPa}$, and 39, 47, or 90 vertical hybrid-pressure levels. The mean tropical troposphere (below 100 $\mathrm{hPa}$) is discretised into 16, 26, or 27 levels, and the mean tropical stratosphere between 100 $\mathrm{hPa}$ and 1 $\mathrm{hPa}$ consists of 15, 15, or 48 levels, respectively.*

- **P5 L20: What do you mean by "Relative to the absolute zonal fluxes..."? Is it the global mean?** Absolute values of zonally averaged fluxes are meant. The sentence has been rephrased accordingly. *Relative to the absolute value of the zonally averaged fluxes, [...]*

- **P5 L30: "Distinct maxima in the seasonal cycle... and minima occurring in late winter." Please rephrase.** The sentence has been rewritten the following: *On both hemispheres, seasonal cycles in zonally averaged VSLS fluxes peak in the summer months and show minima in late winter.*

- **P5 L31: "In case of $CH_2Br_2$, even negative emissions are found during winter at high-latitudes on the northern hemisphere." First, you could explicitly indicate that negative emissions represent a net sink of atmospheric VSLS. Second, how the forcing of LT OH to zero could be affecting this negative flux?** Regarding the first point: It should be indeed mentioned, for it may not be obvious to all readers. Regarding the second point: No net sink for $CH_2Br_2$ in RT1a at the end of the 21st century is found at high latitudes. We have added the following in accordance to the referees suggestions: *Negative emissions representing a net sink of atmospheric $CH_2Br_2$ are found during winter at high-latitudes on the northern hemisphere. [...] Particularly, no net sink for $CH_2Br_2$ occurs at high latitudes in RT1a.*

- **P6 Table 2 caption: "Average absolute flux for year 2000 [...]** The caption has been changed accordingly.

- **P6 L11: I suggest indicating also the absolute increase in VSLS surface VMR.** We follow the suggestion and have added: *Surface values of VSLS increase by 0.47 $\mathrm{ppt}$ (9%), while in the lower stratosphere [...]*

- **P5 L15 and P6 L14: The dependence of the emission flux on wind speed is not explicitly mentioned in Eq. 1.** A detailed description of the wind speed dependence with respect to the chosen $k_{\mathrm{w}}$ parametrization has been added to Section 2. At this point we refer to the given explanation and paper by Lennartz et al. (2015): $k_{\mathrm{w}}$ *is a polynomial function of wind speed depending on the chosen parametrization as mentioned in Section 2.*

- **P6 L17: "Much stronger fluxes have been found in the former simulation in comparison to the latter." First, could you rephrase to make it clear which is the former and which the latter. Second, could the flux difference be due to the different OH zeroing treatment between experiments within the MBL and LT?** Regarding the first point, for clarification, we have changed the phrasing: *Much stronger fluxes (1.3–1.5 times) have been found in RT1a in comparison to SC_free.* Regarding the second point, "zeroing" the [OH] in the LT/MBL enhances the lifetime of VSLS (mainly $CH_2Br_2$), therefore the atmospheric concentrations are higher and the ocean–atmosphere flux is suppressed. We add: *Since OH concentrations in the lower troposphere have been set to zero in SC_free, the atmospheric lifetime and the resulting abundance of VSLS in the lower troposphere is enhanced. Therefore, the total ocean–atmosphere flux is suppressed.*

- **P8 Fig 3: How did you set the $C_{\mathrm{w}}$ from the Ziska et al. (2013) paper for regions in the Arctic that where covered by sea-ice at present time but are not longer covered in the future?** The water concentrations used for the simulation (from Ziska et al. (2013)), do not take ice cover into account. As can be seen exemplarily for $CHBr_3$ in Fig. 1, water concentrations have been extrapolated by Ziska et al. (2013) for regions typically covered by ice at present. In the AIRSEA module, if ice cover (fraction of grid box) is larger than 0.5 the total transfer velocity ($k_{\mathrm{w}}$) is equal to zero else $k_{\mathrm{w}}$ is scaled depending on the fraction of sea ice cover.

[Figure]

**Figure 1.** $CHBr_3$ water concentration in $\mathrm{pmol\,l^{-1}}$ from Ziska et al. (2013)

- P12 L18 and elsewhere in the text: is *ansatz* an accepted English word?: *Ansatz* is indeed an accepted word in English. See https://en.oxforddictionaries.com/definition/ansatz for details.

- P13 Fig.6: Have you thought about including the $CH_2Br_2$ and $CHBr_3$ photolysis rate vertical profile in a second panel? An early version of the manuscript indeed included a figure of photolysis rate vertical profiles. We have put a figure into the revised manuscript.

- P13 L11: "At about $20\,\mathrm{hPa}$..." Do you mean a $20\,\mathrm{hPa}$ difference from the mean tropopause? Without looking at the corresponding figure, the reference to $20\,\mathrm{hPa}$ at this point is indeed not clear. We have rephrased the sentence to clarify that $20\,\mathrm{hPa}$ is meant with respect to the mean tropopause: *Within $20\,\mathrm{hPa}$ with respect to the mean tropopause, [...].*

- Regarding Fig. 9, and considering P8 L1 "There is an upward shift of the tropopause height of about $8\,\mathrm{hPa}$ between present day and future". Why would you show such a large shift in pressure respect to the mean tropopause ($\pm100\,\mathrm{hPa}$) if the difference in the tropopause pressure is smaller than $10\,\mathrm{hPa}$? I would expect the changes in the tropopause height to affect only the UTLS, and partitioning between SG and PG, but not the overall total bromine abundances in the middle and upper stratosphere. There are two aspects which we are addressing in Section 4 of our paper. As others have also shown, the future troposphere is warming and enlarging, while the stratosphere is cooling and shrinking. In accordance, the tropopause is rising. We also see an enhancement of vertical tracer transport in tropical regions, pushing younger air further upwards. This air has still a higher amount of SG which have not yet degenerated to PG. This physical effect is in fact shown in Fig. 6 and discussed in the beginning of Section 4. In Figure 10, which is discussed in Section 4.2, we look at a different aspect. Due to the physical changes, accompanied by a rise of the mean tropopause in the tropics, it is not entirely valid to simply compare VMR at the "same" pressure level in the UTLS between present and future. The entire profile is shifted upwards not only the UTLS region, although the UTLS region includes the most extreme example: "[A]ir which at present is considered stratospheric will be still tropospheric in the future." We make this point clearer in the final manuscript version.

- P15 L14: If the authors are willing to address the impact of VSLS in the future evolution of Antarctic ozone, they should at least compare their results respect to Oman et al. (2016) and Fernandez et al. (2017). We are not in detail addressing changes in the Antarctic ozone hole, but we will include the references and compare our results to these papers.

- P18 L29: "... and aerosol formation have been taken into consideration." While a full aerosol treatment has been considered for some of the simulations, the sentence gives the impression that an aerosol formation module for VSLS has been considered in this work. I suggest rephrasing to avoid misleading interpretations. Indeed, only a small portion of the entire paper is based on simulations including an aerosol formation treatment (RT1a/RT1b). RT1a/RT1b are only analysed in the context of VSLS influence on future ozone. We do not study aerosols explicitly in this paper and have in fact no special formation mechanism of aerosols involving VSLS. We rewrite the sentence to prevent from misleading interpretations: *
[revised manuscript text omitted]